# Marine viruses disperse bidirectionally along the natural water cycle

Janina Rahlff [1,2,8] ✉, Sarah P. Esser [1,3], Julia Plewka [1,3], Mara Elena Heinrichs[4], André Soares [1,3], Claudio Scarchilli [5], Paolo Grigioni [5], Heike Wex [6], Helge-Ansgar Giebel [4,9] & Alexander J. Probst [1,3,7]

Marine viruses in seawater have frequently been studied, yet their dispersal from neuston ecosystems at the air-sea interface towards the atmosphere remains a knowledge gap. Here, we show that 6.2% of the studied virus population were shared between air-sea interface ecosystems and rainwater. Virus enrichment in the 1-mm thin surface microlayer and sea foams happened selectively, and variant analysis proved virus transfer to aerosols collected at ~2 m height above sea level and rain. Viruses detected in rain and these aerosols showed a significantly higher percent G/C base content compared to marine viruses. CRISPR spacer matches of marine prokaryotes to foreign viruses from rainwater prove regular virus-host encounters at the air-sea interface. Our findings on aerosolization, adaptations, and dispersal support transmission of viruses along the natural water cycle.

Marine viruses represent the most abundant biological entities in the oceanic water column[1] where they contribute to microbial diversity[2], can influence host metabolism by providing auxiliary metabolic genes[3], and influence carbon cycling by inducing host cell lysis (the viral shunt)[4] (reviewed by Mateus[5]). While viruses have been studied in many marine ecosystems including surface waters[6] and deep-sea sediments[7], their presence at the air-sea interface, where microorganisms modulate gas and organic matter exchange processes[8–10], remains mostly enigmatic. Like many micro- and macroorganisms[11,12], viruses accumulate in the thin (<1 mm) uppermost layer of aquatic ecosystems, the surface microlayer (SML, reviewed by Cunliffe, et al.[13], Engel, et al.[14]), where they belong to a pool of organisms collectively referred to as neuston[15]. The enrichment of the virioneuston in the SML is mediated by bubble transport from the

underlying water[16,17] and likely maintained by viral attachment to particles[18] as well as a dependency on abundant prokaryotic hosts[19]. In freshwater, bacteriophages residing in the SML can form autochthonous communities[20] but comparatively little (viral) metagenomics studies have been conducted for marine SML (reviewed by Rahlff[21]). More recently, large-scale sampling efforts during the Tara Pacific expedition and subsequent amplicon sequencing provided insights into the surface ocean and aerosolized bacteria, their diversity, and their potential sources[22,23], but such insights are lacking for viruses.

Sea foams float as (extended) patches on the sea surface (Supplementary Movie 1), forming deposits at the shoreline and being microbial habitats that contrast the SML in terms of their microbial community composition[24,25]. Based on satellite data, foams

[1]Group for Aquatic Microbial Ecology, Department of Chemistry, Environmental Microbiology and Biotechnology (EMB), University of Duisburg-Essen, 45141 Essen, Germany. [2]Centre for Ecology and Evolution in Microbial Model Systems (EEMiS), Department of Biology and Environmental Science, Linnaeus University, 39231 Kalmar, Sweden. [3]Environmental Metagenomics, Research Center One Health Ruhr of the University Alliance Ruhr, University of Duisburg-Essen, 45141 Essen, Germany. [4]Institute for Chemistry and Biology of the Marine Environment (ICBM), Carl von Ossietzky University of Oldenburg, 26129 Oldenburg, Germany. [5]Italian National Agency for New Technologies, Energy and Sustainable Economic Development (ENEA), 00123 Rome, Italy. [6]Atmospheric Microphysics, Leibniz Institute for Tropospheric Research (TROPOS), 04318 Leipzig, Germany. [7]Centre of Water and Environmental Research (ZWU), University of Duisburg-Essen, 45141 Essen, Germany. [8]Present address: Aero-Aquatic Virus Research Group, Faculty of Mathematics and Computer Science, Friedrich Schiller University Jena, 07743 Jena, Germany. [9]Present address: Institute for Chemistry and Biology of the Marine Environment (ICBM), Center for Marine Sensors (ZfMarS), Carl von Ossietzky University of Oldenburg, 26382 Wilhelmshaven, Germany. ✉e-mail: Janina.rahlff@uol.de

(whitecaps) cover up to 6% of oceanic surface area and are expected to become more frequent with climate change[26]. During storms, foams can flood beaches[27] and massively pollute coastal areas like recently in Turkey[28]. Furthermore, sea foams can contain pathogenic bacteria[29] and their easy spread might be an important step for the dispersal and aerosolization of its inhabiting microbes[30] and potentially viruses. Foams can effectively concentrate viruses[31] which survive more than three hours of drying and sunlight when caught in foams[16]. Virus-like particles (VLPs) can reach a 300 × higher abundance in foams compared to surrounding waters[28].

Interest in studying viruses in the skin-like layer between ocean and atmosphere arises from therein appearing human pathogenic viruses[32], the potential of SML viruses to get airborne[33], to selectively enrich in aerosols[34,35], and to disperse over long distances to eventually promote turnover of algal blooms in remote regions[36]. A recent review highlighted the need to quantify marine aerobiota, to characterize the spatial-temporal dimensions of dispersal, and to understand the acclimations of marine microorganisms to atmospheric conditions[37]. Virus aerosolization from the SML was previously studied[16,33,35], but investigations pursuing metagenomic approaches to explore the virioneuston and its aerosolization in the field are lacking.

Once airborne, viruses could even fulfill other functions as recently suggested[38]: Airborne marine viruses could serve as ice-nucleating particles (INPs), a function already described for many microorganisms[39,40], and act as catalysts to mediate freezing at temperatures warmer than −10 °C. INPs exist in the SML[41-43] and has an important role in cloud formation, cloud albedo, and precipitation and thus are key in climate regulation dynamics[44]. Viruses and bacteria can be found in clouds[45-48], where the latter might grow selectively[49] and as INPs, trigger their own precipitation[50,51]. Precipitation could be an underestimated source of microorganisms to Earth's surface, for example, it contributed to as much as 95% of atmospheric bacterial deposition at a Korean site[49]. So far, research on viruses included in wet precipitation was mainly focused on viruses relevant to human health, such as enteric or adenoviruses[52-54]. Reche, et al.[55] reported that $10^7$ bacteria and $10^9$ viruses deposit from the atmosphere per m² per day, with marine sources having stronger contributions than terrestrial ones. This rate can be one order of magnitude higher for bacteria[56] and perhaps also for viruses. Rain events related to a hurricane decreased marine viral diversity and abundance as well as introduced new taxa in the western Gulf of Mexico[57]. Furthermore, stormwater runoff changed the viral community composition of inland freshwaters and stormwater retention pond[58,59]. However, the transmission cycle of neustonic viruses from the sea surface via aerosols to wet precipitation remains speculative and the degree of viral exchange between these ecosystems and along the natural water cycle is unknown.

To address some of these knowledge gaps, we analyzed 55 metagenomes including samples from the air-sea interface (sea foams and SML), subsurface water (SSW) from 1 m depth, aerosols in the boundary layer collected from ~2 m above the sea level, and precipitation (rain, snow) collected in a coastal region of the Skagerrak in Tjärnö, Sweden (Fig. 1a). We explored the potential of marine viruses to become aerosolized and being returned to Earth via wet deposition. Here, we show that viruses become aerosolized from the sea surface, with certain marine viruses being detected in rainwater. Rainfall connected to air masses that had prolonged exposure to the ocean is associated with a higher prevalence of marine viruses and genomes of marine prokaryotes. Viruses exclusively found in rainwater and those from boundary layer aerosols exhibit a significantly higher G/C base content in their genome compared to marine viruses. Furthermore, our analysis of virus-host relationships, using clustered regularly interspaced short palindromic repeats (CRISPR) systems, reveals connections between viruses and hosts across ecosystem boundaries. We conclude that marine viruses travel bidirectionally along the natural water cycle.

## Results

### Cell and VLP abundance, enrichments, and their correlations reveal tight virus-host associations in the neuston

Marine-, aerosol-, and rain samples were collected around Tjärnö Marine Laboratory, Sweden including eleven stations in coastal waters of the Skagerrak, where air-sea interface samples (SML, foam) and a reference depth were sampled (Fig. 1a). Prokaryotic, small phototrophic eukaryotic, and VLP counts were measured to assess virus-host ratios and correlations in the neuston (SML, foam) compared to the underlying plankton in the SSW. Across all stations, viral abundance ranged between $5.0 \times 10^7$–$1.8 \times 10^8$, $1.3 \times 10^7$–$3.4 \times 10^7$, and $1.4 \times 10^7$–$2.0 \times 10^7$ VLPs mL⁻¹ in floating sea foams, the SML, and SSW, respectively, supporting a VLP gradient towards the atmosphere (Table 1, Supplementary Fig. 1). Numbers of VLPs were verified by microscopic analysis as shown representatively for Station 4 (Fig. 1b–d, Supplementary Fig. 2), and the images revealed that VLPs in sea foams often adhered to particulate matter (Fig. 1e). Counts of VLP in precipitation samples (rainwater) ranged between $3.7 \times 10^4$–$3.4 \times 10^5$ VLPs mL⁻¹. Enrichment factors (EF) for VLPs in the SML over SSW varied between 0.7 (depletion) and 1.9 (enrichment; Table 1). Total cell numbers of prokaryotes were $1.3 \times 10^6$–$3.8 \times 10^6$, $7.0 \times 10^5$–$1.1 \times 10^6$ as well as $7.0 \times 10^5$–$8.7 \times 10^5$ cells mL⁻¹ in floating sea foams (representative aggregations in Fig. 1f), the SML, and SSW, respectively (Supplementary Fig. 1, Table 1). EFs for prokaryotes fluctuated between 0.9 and 1.3. Across the five precipitation samples, $2.7 \times 10^3$–$1.8 \times 10^4$ prokaryotic cells mL⁻¹ were detected. Virus-host ratios (host = prokaryotes) based on flow cytometry data were highest in foams (range = 25.3–48.4), followed by the SML (range = 15.5–34.2) and SSW (range = 19.3–26.7). Virus-host ratios in precipitation samples showed the strongest variation and ranged between 7.1 and 127.8 (Table 1). The highest virus-host ratios in the SML were detected on days were VLP EFs were ≥1.8 and prokaryotic EFs ≥1.1 at the same time. Total cell numbers of small phototrophic eukaryotes ranged between $3.4 \times 10^3$–$1.8 \times 10^4$, $1.8 \times 10^3$–$6.2 \times 10^3$, $2.1 \times 10^3$–$5.0 \times 10^3$ cells mL⁻¹ in sea foams, SML, and SSW, respectively.

Within the SML, the number of small phototrophic eukaryotes and prokaryotes was significantly correlated with VLP abundance (Pearson's corr = 0.74, t = 3.13, p = 0.014, df = 8, n = 10 and Pearson's corr = 0.70, t = 2.75, p = 0.025, df = 8, n = 10, respectively, Fig. 2a, b), while the plankton/SSW correlations with the same variables were not significant (Table 2, Fig. 2c, d). In addition, absolute numbers of small phototrophic eukaryotes and their EFs were significantly positively correlated with absolute numbers and EF of prokaryotes for the neuston, inferring a common transfer mechanism of these cell types towards the air-sea interface (Table 2, Supplementary Fig. 3). EFs of VLPs were significantly correlated to EFs of prokaryotes (Spearman's rho = 0.83, p = 0.006, n = 10) but not to EFs of small phototrophic eukaryotes (Fig. 2e, f) probably indicating that enrichments of viruses in the SML are dependent on host cell availability and that most SML viruses are prokaryotic viruses.

### Cell counts in the SSW but not the SML show significant correlations with salinity

Absolute VLP counts derived from neuston and SSW plankton as well as EFs for VLPs and cells correlated with environmental data (light, salinity, and wind speed) did not reveal any significant relationships (Table 2). However, numbers of small phototrophic eukaryotes (Pearson's corr = 0.68, t = 2.59, p = 0.031, df = 8, n = 10) and prokaryotes (Pearson's corr = 0.71, t = 2.83, p = 0.022, df = 8, n = 10) from the SSW but not the SML were significantly correlated with salinity, which could

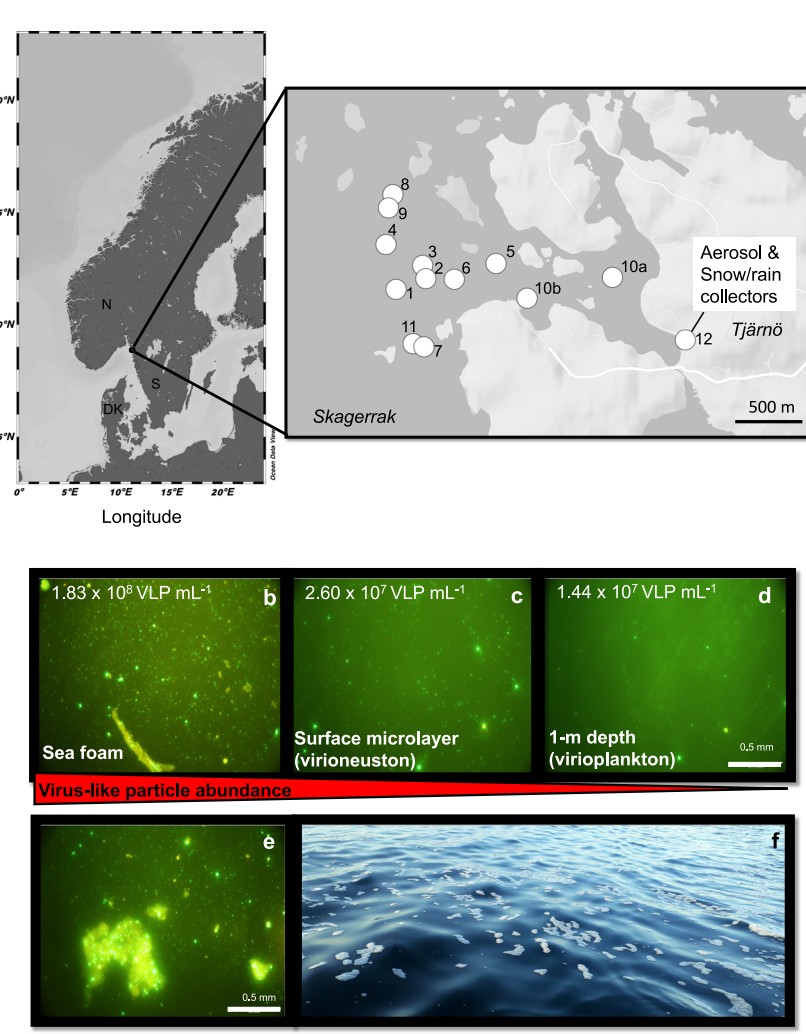

**Fig. 1 | Map depicting sampling stations and viral enrichment in SML and sea foam.** Map of sampling sites was generated using Ocean Data View[143] and Map Maker https://maps.co. For further details about the stations, please refer to Supplementary data 9 (**a**). Gradient of virus-like particles (VLP) in sea foam (**b**), surface microlayer (**c**), and 1-m deep subsurface water (**d**) recorded in epifluorescence microscopy with VLP counts mL$^{-1}$ from Station 4 as obtained from flow cytometry. VLPs stick to particulate matter in foams (**e**). Sea foams were collected as floating patches from the ocean's surface (representative photograph, (**f**). Source data are provided as a Source Data file.

be explained by regular inflow of high saline water from the Atlantic Ocean that probably affects deeper water layers more than the SML.

We applied linear models to investigate combinatory effects of environmental variables (wind speed, light, and salinity) on *EF* of cells and VLPs in the SML. One linear model considering the combinatory effects of wind speed and salinity on the *EF* of small phototrophic eukaryotes in the SML was significant (F-test, $F = 5.43$, $p = 0.038$, $df = 6$), and in total 59.6% of the residuals could be explained by this model. This could indicate that due to their bigger sizes, the enrichment of phototrophic eukaryotes in the SML is more affected by wind and currents than that of the prokaryotes. The model's Akaike information criterion (*AIC*) was −2.37, which was superior to considering wind speed (*AIC* = 5.84) and salinity (*AIC* = 6.40) alone. Other models testing single and combined environmental parameters on the *EF* of cells and VLP in the SML were not significant.

## Ice nucleation activity of marine samples was highest in sea foams

The highest ice nucleation activity concluded from INP concentrations over the detectable temperature range in our samples was determined for sea foams, followed by SML and SSW samples (Supplementary Fig. 4). Ice nucleation activity for all samples generally started at high

temperatures of ·−4 to −6 °C, comparable to observations for microorganisms in the atmosphere[60].

## Aerosolization of biota and decreasing diversity from marine ecosystems towards the boundary layer

We found a lower diversity of rain and aerosol microbiota compared to marine samples, with the difference between aerosols sampled from the boundary layer and SSW being significant (Kruskal-Wallis with Dunn's multiple comparisons test, $p = 0.0136$, Fig. 3a). Significant differences for beta-diversity were detected between ecosystems ($p < 0.001$, Fig. 3b), namely between aerosols and foam (TukeyHSD, $p = 0.02$), aerosols and SML ($p = 0.02$), and aerosols and SSW ($p = 0.0002$, Supplementary Fig. 5a). Furthermore, aerosol and rain communities were distinct from marine communities and from each other (Supplementary Fig. 6a, Supplementary data 13). SSW samples were mainly composed of Proteobacteria (mean relative abundance ± standard deviation = $67.8 \pm 5.2\%$, $n = 9$), Bacteroidetes ($22.6 \pm 6.0\%$), and Thaumarchaeota ($5.1 \pm 2.3\%$, Fig. 3c). In general, the SML samples reflected this composition, although two samples deviated by containing a major percentage of Planctomycetes ($13.3 \pm 28.7\%$, $n = 9$) and Cyanobacteria ($4.6 \pm 7.2\%$, $n = 9$). Sea foams were like the SML but additionally contained WOR-2 ($2.7 \pm 3.2\%$, $n = 3$)

**Table 1 | Cell and virus-like particle (VLP) abundances for precipitation (PRC), foam, surface microlayer (SML) and subsurface water (SSW) samples**

| St | Date | Viral abundance (VLP mL⁻¹) | | | | | Prokaryote abundance (cells mL⁻¹) | | | | | Eukaryote abundance (cells mL⁻¹) | | | | Virus to prok. host ratio | | | |
|---|---|---|---|---|---|---|---|---|---|---|---|---|---|---|---|---|---|---|---|
| | | Foam | SML | SSW | PRC | EF (SML/SSW) | Foam | SML | SSW | PRC | EF (SML/SSW) | Foam | SML | SSW | EF (SML/SSW) | Foam | SML | SSW | PRC |
| 1 | 3. Feb. 20 | n.d. | 1.8E+07 | 1.9E+07 | n.d. | 0.9 | n.d. | 7.9E+05 | 8.0E+05 | n.d. | 1.0 | n.d. | 3.3E+03 | 3.4E+03 | 1.0 | n.d. | 22.4 | 23.5 | n.d |
| 2 | 4. Feb. 20 | n.d. | 1.7E+07 | 1.7E+07 | n.d. | 1.0 | n.d. | 8.5E+05 | 8.4E+05 | n.d. | 1.0 | n.d. | 4.5E+03 | 4.3E+03 | 1.0 | n.d. | 19.4 | 20.0 | n.d |
| 3 | 6. Feb. 20 | n.d. | 1.7E+07 | 1.6E+07 | n.d. | 1.0 | n.d. | 7.9E+05 | 7.9E+05 | n.d. | 1.0 | n.d. | 2.1E+03 | 2.1E+03 | 1.0 | n.d. | 21.2 | 20.4 | n.d |
| 4 | 7. Feb. 20 | 1.8E+08 | 2.6E+07 | 1.4E+07 | n.d. | 1.8 | 3.8E+06 | 8.0E+05 | 7.5E+05 | n.d. | 1.1 | 1.8E+04 | 3.4E+03 | 3.4E+03 | 1.0 | 48.4 | 32.5 | 19.3 | n.d |
| 12/R | 10. Feb. 20 | n.d. | n.d. | n.d. | 6.5E+04 | n.d. | n.d. | n.d. | n.d. | 9.1E+03 | n.d. | n.d. | n.d. | n.d. | n.d. | n.d. | n.d. | n.d. | 7.1 |
| 5 | 11. Feb. 20 | n.d. | 1.3E+07 | 1.9E+07 | n.d. | 0.7 | n.d. | 8.5E+05 | 8.7E+05 | n.d. | 1.0 | n.d. | 3.2E+03 | 5.0E+03 | 0.6 | n.d. | 15.5 | 21.7 | n.d |
| 6 | 13. Feb. 20 | 5.0E+07 | 1.6E+07 | 1.9E+07 | n.d. | 0.9 | 1.3E+06 | 8.0E+05 | 7.8E+05 | n.d. | 1.0 | 3.4E+03 | 3.5E+03 | 3.0E+03 | 1.2 | 39.4 | 20.2 | 24.4 | n.d |
| 7 | 14. Feb. 20 | n.d. | 1.9E+07 | 2.0E+07 | n.d. | 0.9 | n.d. | 7.0E+05 | 7.5E+05 | n.d. | 0.9 | n.d. | 1.8E+03 | 3.4E+03 | 0.5 | n.d. | 26.8 | 26.7 | n.d |
| 8 | 15. Feb. 20 | 7.7E+07 | 3.4E+07 | 1.9E+07 | n.d. | 1.8 | 3.0E+06 | 9.9E+05 | 7.6E+05 | n.d. | 1.3 | 1.8E+04 | 6.2E+03 | 4.2E+03 | 1.5 | 25.3 | 34.2 | 24.4 | n.d |
| 12/R | 16. Feb. 20 | n.d. | n.d. | n.d. | 3.7E+04 | n.d. | n.d. | n.d. | n.d. | 3.9E+03 | n.d. | n.d. | n.d. | n.d. | n.d. | n.d. | n.d. | n.d. | 9.5 |
| 9 | 19. Feb. 20 | n.d. | 3.1E+07 | 1.6E+07 | n.d. | 1.9 | n.d. | 1.1E+06 | 8.3E+05 | n.d. | 1.3 | n.d. | 5.6E+03 | 4.5E+03 | 1.2 | n.d. | 28.1 | 19.8 | n.d |
| 12/R | 21. Feb. 20 | n.d. | n.d. | n.d. | 1.2E+05 | n.d. | n.d. | n.d. | n.d. | 7.9E+03 | n.d. | n.d. | n.d. | n.d. | n.d. | n.d. | n.d. | n.d. | 14.9 |
| 12/R | 22. Feb. 20 | n.d. | n.d. | n.d. | 2.7E+05 | n.d. | n.d. | n.d. | n.d. | 1.8E+04 | n.d. | n.d. | n.d. | n.d. | n.d. | n.d. | n.d. | n.d. | 14.8 |
| 10 | 24. Feb. 20 | 5.5E+07 | 2.0E+07 | 1.9E+07 | n.d. | 1.1 | 1.5E+06 | 9.2E+05 | 8.5E+05 | n.d. | 1.1 | 6.1E+03 | 3.9E+03 | 3.9E+03 | 1.0 | 36.2 | 21.9 | 22.1 | n.d |
| 11 | 26. Feb. 20 | n.d. | n.d. | 1.5E+07 | n.d. | n.d. | n.d. | n.d. | 7.0E+05 | n.d. | n.d. | n.d. | n.d. | 2.1E+03 | n.d. | n.d. | n.d. | 21.6 | n.d |
| 12/R | 26. Feb. 20 | n.d. | n.d. | n.d. | 3.4E+05 | n.d. | n.d. | n.d. | n.d. | 2.7E+03 | n.d. | n.d. | n.d. | n.d. | n.d. | n.d. | n.d. | n.d. | 127.8 |

Enrichment factor (EF) for VLPs, small phototrophic eukaryote and prokaryote (prok.) enrichment in the SML over SSW as described in the main text as well as VLP to prokaryotic host ratios were calculated; n.d. = not determined, "R" under station refers to precipitation sampling, St = Station. Small phototrophic eukaryotes in precipitation were measured but could not be detected. PRC on the 26th Feb. 2020 was a mixture of snow and rain.

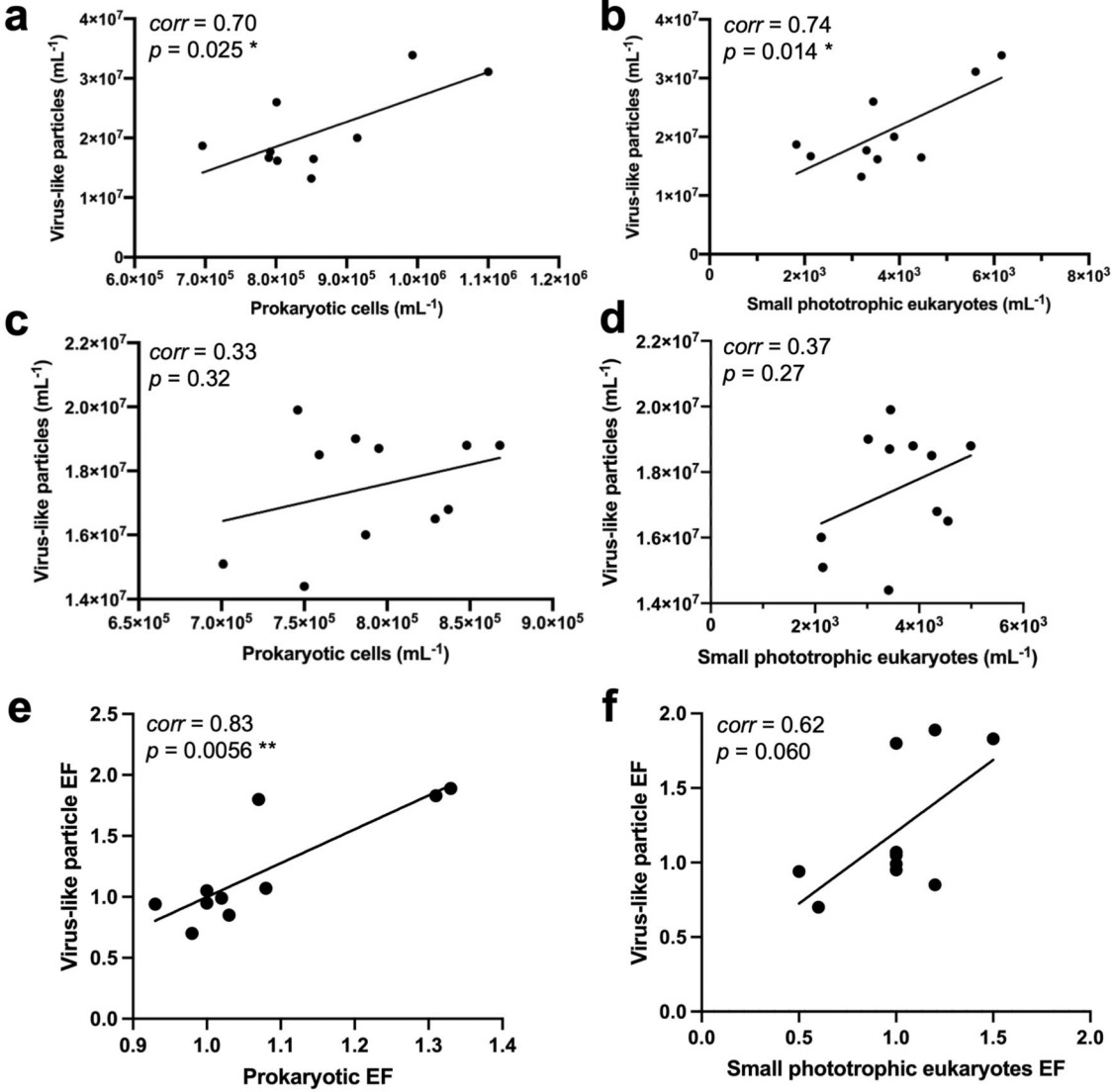

**Fig. 2 | Relationship of virus-like particles (VLPs) and host cells in the neuston and the plankton.** Linear regression for VLPs versus prokaryotic cells (**a**) and small phototrophic eukaryotes (**b**) in the surface microlayer corresponding to the neuston. Linear regression for VLPs versus prokaryotic cells (**c**) and small phototrophic eukaryotes (**d**) in the subsurface water corresponding to the plankton.

Linear regression for VLP enrichment factors (*EF*) versus prokaryotic *EF* (**e**) and small phototrophic eukaryotes *EF* (**f**) for the surface microlayer compared to subsurface water. The correlation coefficient (*corr*) and *p*-value for the correlations are shown in the figure, and further information on the statistics is provided in Table 2. *$p \leq 0.05$; **$p \leq 0.01$. Source data are provided as a Source Data file.

and an increasing proportion of Bacteroidetes (37.7 ± 11.6%, $n = 3$). Aerosols also contained Proteobacteria (43.4 ± 13.9%, $n = 8$), Bacteroidetes (21.0 ± 21.6%), and Planctomycetes (20.8 ± 13.0%). The snow sample contained a relative abundance of 97.3% Cyanobacteria (*Rivularia* sp.). The mean relative abundances of Proteobacteria (49.9 ± 5.1%) and Bacteroidetes (20.7 ± 2.4%) in rain were comparable to those in aerosols, but in contrast to sea surface water and aerosols, Actinobacteria (6.3 ± 9.4%) and Cyanobacteria (13.6 ± 5.1%) were more abundant (Fig. 3c).

Detection of the same bacterial ribosomal protein S3 (*rpS3*) genes in marine, aerosol and precipitation samples suggests their aerosolization from the sea surface into the boundary layer, e.g., for Proteobacteria (*Oceanospirillum maris*, *Loktanella vestfoldensis*, *Candidatus* Pelagibacter), Bacteroidetes (*Crocinitomix catalasitica* and *Bacteroides fragilis*), Cyanobacteria (*Crinalium epipsammum* and *Oscillatoria* sp.) and Planctomycetes (*Pirellula staleyi*, Fig. 3c, d). Aerosols and precipitation contained Cyanobacteria such as *Rivularia* sp. (max. 14.1% in rain, 97.7% in snow) and Proteobacteria such as

*Sphingobium japonicum* (max. 21.8%) or *Methyloferula stellata* (max. 21.4%), which could not be found in any of the local marine samples. Rain contained Actinobacteria (*Cryocola* sp., max. 20.3%) and Bacteroidetes such as *Mucilaginibacter paludis* (max. 20.1%) that were only scarcely detected in marine samples based on relative abundance (<0.2%).

**K-mer based virus-host assignments reveal marine *Pelagibacter* and *Porticoccus* from boundary layer and rain as prevalent hosts**
In total, 116 metagenome-assembled genomes (MAGs) could be recovered from 24 different samples (Supplementary Data 1 and 2), which ranged from 58.8–100% completeness (median = 86.3%) and 0–11.8% contamination (median = 3.9%) based on quality criteria implemented in uBin[61]. CheckM[62] resulted in completeness and contamination scores of 18.4–99.5% (median = 80.6%) and 0–17.3% (median = 1.6%) for these MAGs, respectively. Most host MAGs were of bacterial origin, except for three assigned to the genus *Nitrosopumilus* (Archaea). Recovering MAGs from non-marine samples resulted in a

**Table 2 | Statistical results for correlations (two-sided tests) and linear models**

| Correlation analysis | X | Y | Test | t value | df | p value | corr | Significance |
|---|---|---|---|---|---|---|---|---|
| Absolute counts | Prok_SML | VLP_SML | Pearson | 2.76 | 8 | 0.025 | 0.7 | * |
| | Euk_SML | VLP_SML | Pearson | 3.13 | 8 | 0.014 | 0.74 | * |
| | Prok_SSW | VLP_SSW | Pearson | 1.02 | 9 | 0.32 | 0.33 | n.s. |
| | Euk_SSW | VLP_SSW | Pearson | 1.18 | 9 | 0.27 | 0.37 | n.s. |
| | Euk_SML | Prok_SML | Pearson | 5.32 | 8 | 7.10E-04 | 0.88 | *** |
| | Euk_SSW | Prok_SSW | Pearson | 2.95 | 9 | 0.016 | 0.7 | * |
| Enrichment factors (SML/SSW) | EF_Prok | EF_VLP | Spearman | | | 5.60E-03 | 0.83 | ** |
| | EF_Euk | EF_VLP | Spearman | | | 0.06 | 0.62 | n.s. |
| | EF_Euk | EF_Prok | Spearman | | | 5.50E-03 | 0.83 | ** |
| Env.variables_vs_absolute_counts | Wind speed | VLP_SML | Pearson | 0.36 | 8 | 0.73 | 0.13 | n.s. |
| | | VLP_SSW | Pearson | 0.22 | 8 | 0.83 | 0.078 | n.s. |
| | | Prok_SML | Pearson | 1.72 | 8 | 0.12 | 0.52 | n.s. |
| | | Prok_SSW | Pearson | 1.93 | 8 | 0.09 | 0.56 | n.s. |
| | | Euk_SML | Pearson | 1.87 | 8 | 0.10 | 0.55 | n.s. |
| | | Euk_SSW | Pearson | 2.38 | 8 | 0.044 | 0.64 | * |
| | Salinity | VLP_SML | Pearson | 0.26 | 8 | 0.8 | 0.092 | n.s. |
| | | VLP_SSW | Pearson | −1.11 | 8 | 0.3 | −0.37 | n.s. |
| | | Prok_SML | Pearson | 1.84 | 8 | 0.1 | 0.54 | n.s. |
| | | Prok_SSW | Pearson | 2.83 | 8 | 0.022 | 0.71 | * |
| | | Euk_SML | Pearson | 1.63 | 8 | 0.14 | 0.5 | n.s. |
| | | Euk_SSW | Pearson | 2.59 | 8 | 0.032 | 0.68 | * |
| | Light | VLP_SML | Pearson | −1.27 | 8 | 0.24 | −0.41 | n.s. |
| | | VLP_SSW | Pearson | 0.24 | 8 | 0.82 | 0.084 | n.s. |
| | | Prok_SML | Pearson | 0.03 | 8 | 0.98 | 0.009 | n.s. |
| | | Prok_SSW | Pearson | 1.62 | 8 | 0.14 | 0.5 | n.s. |
| | | Euk_SML | Pearson | −0.84 | 8 | 0.43 | −0.28 | n.s. |
| | | Euk_SSW | Pearson | −0.75 | 8 | 0.48 | −0.26 | n.s. |
| Env.variables_vs_enrichment | Wind speed | EF_VLP | Spearman | | | 0.81 | −0.09 | n.s. |
| | | EF_Prok | Spearman | | | 0.30 | 0.36 | n.s. |
| | | EF_Euk | Spearman | | | 0.47 | 0.26 | n.s. |
| | Salinity | EF_VLP | Spearman | | | 0.47 | 0.26 | n.s. |
| | | EF_Prok | Spearman | | | 0.39 | 0.31 | n.s. |
| | | EF_Euk | Spearman | | | 0.41 | 0.30 | n.s. |
| | Light | EF_VLP | Spearman | | | 0.81 | −0.09 | n.s. |
| | | EF_Prok | Spearman | | | 0.89 | −0.05 | n.s. |
| | | EF_Euk | Spearman | | | 0.33 | −0.35 | n.s. |
| Linear model | X | Y | Adjusted R² | F value | df | p value | AIC | Significance |
| | Wind*Salinity | EF_Euk | 0.596 | 5.43 | 6 | 0.038 | −2.377 | * |

*AIC* Akaike's Information Criterion, *corr* correlation coefficient, *df* degrees of freedom, *EF* enrichment factor, *Euk* small phototrophic eukaryotes, *Prok* prokaryotes, *SML* surface microlayer, *SSW* subsurface water (1 m depth), *VLP* virus-like particles, significance levels: *<0.05, **<0.01, ***<0.001, *n.s.* not significant. Only linear models with significance in *F*-test are shown. Tested were all possible combinations of environmental variables (light, wind speed, salinity) on VLP and cell enrichment in the SML.

single MAG obtained from rain (genus *Pedobacter*) and one from an aerosol sample (class Planctomycetes, order Pirellulales), respectively. Overall, bacterial MAGs were mostly classified as Gammaproteobacteria ($n = 43$), Alphaproteobacteria ($n = 30$), Bacteroidia ($n = 36$), and Planctomycetes ($n = 4$). Based on read mapping and breadth, all MAGs were detected in a marine ecosystem (except for the *Pedobacter* sp. MAG), rain and some additionally in boundary layer aerosols (Supplementary data 3). MAGs were matched to viruses based on shared *k*-mer frequency patterns, revealing that 120 marine viruses matched a MAG assigned to *Candidatus* Pelagibacter (Supplementary Fig. 7). Hosts of rain viruses (not detectable in the other sampled ecosystems) and one aerosol virus were predicted MAGs belonging to the family Porticoccaceae and Flavobacteriaceae.

## Viral diversity and transfer from the sea surface to aerosols and rain

Alpha-diversity was significantly different for viruses between boundary layer aerosols and SSW (Dunn's multiple comparison test, $p < 0.0001$), but also different between SML > 0.2 μm fraction and SSW virome samples ($p = 0.029$, Fig. 4a). The distinct viral community of SML > 0.2 μm samples was also demonstrated by beta-diversity analysis (Fig. 4b, c). Here, significant differences for the NMDS analysis were found ($p = 0.001$), including significant differences between the SSW virome and the SML 0.2 μm fraction (TukeyHSD, $p = 0.013$, Supplementary Fig. 5b). We further investigated on SML and foam viral clusters (VCs) detected in boundary layer aerosols and rain. Rainwater contained the abundant cluster VC_723_0 being absent in samples from the other

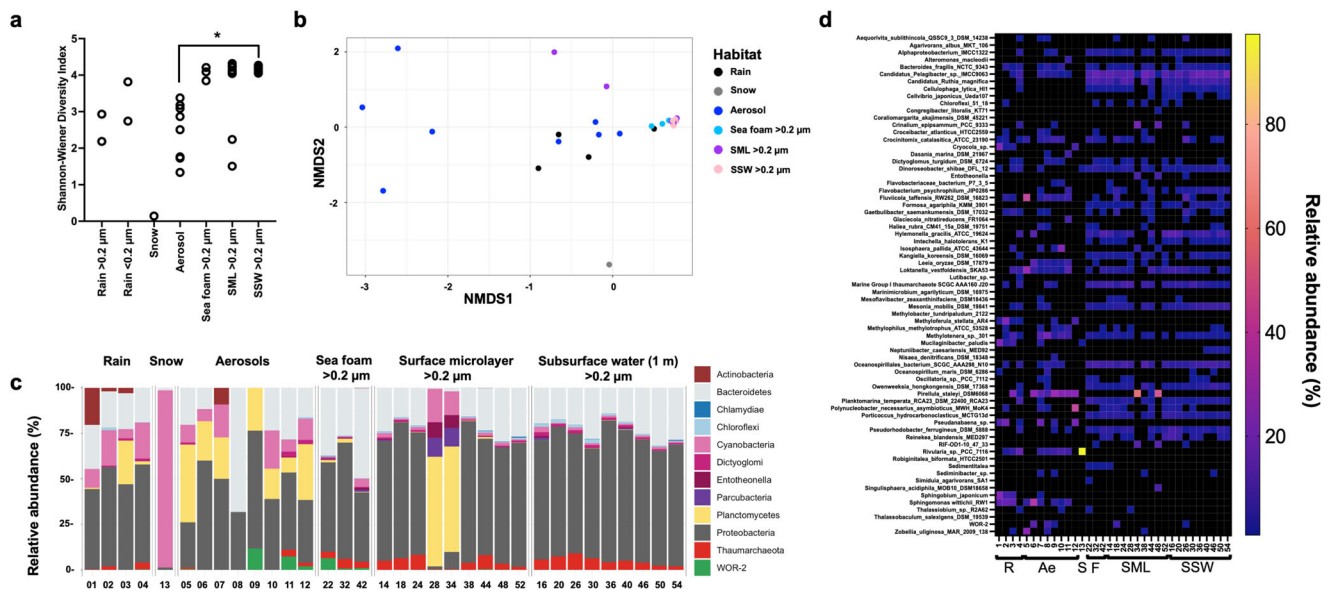

**Fig. 3 | Diversity and relative abundance of marine and airborne prokaryotes based on relative abundance in rain, snow, aerosols, sea foam, surface microlayer (SML), and subsurface water (SSW).** Diversity depicted by Shannon-Wiener index with * = adjusted $p < 0.05$, here adjusted $p = 0.0136$ in Dunn's multiple comparison test (post hoc analysis after Kruskal-Wallis test) for rain >0.2 μm ($n = 2$), rain <0.2 μm ($n = 2$), snow ($n = 1$), aerosol ($n = 8$), sea foam ($n = 3$), SML ($n = 9$), and SSW ($n = 9$) samples (**a**), non-metric multidimensional scaling plot based on Bray-Curtis dissimilarity (stress = 0.082) (**b**), and stacked bar chart on beta diversity at the phylum level (**c**). In (**c** and **d**), relative abundance is based on read-normalized coverage on scaffolds carrying the ribosomal protein S3 gene (*rpsS3*) as explained in the main text. Black fields represent accumulated taxonomic units of <1% relative abundance. Seawater samples show result of >0.2 μm samples, whereas rain contains >0.2 μm (#1 + #3) and viromes (#2 + #4). The heat map shows the relative abundance of all identified prokaryotic taxa across different ecosystems (**d**) (R = rain, S = snow, Ae = aerosols, F = sea foam, SML = surface microlayer, SSW = subsurface water). Sample 13 (snow) contains 97.3% relative abundance of *Rivularia* sp., which is out of the range of the scale and thus not shown. Sample number is in accordance with Supplementary Data 12. Source data are provided as a Source Data file.

ecosystems (Fig. 4c) and had one associated scaffold related to *Rhizobium* phage RHph_N3_2. *EF*s were overall higher for VCs in rain (max. *EF* = 15.8) compared to enrichments in aerosols in reference to SML and foam (max. *EF* = 2.8, Fig. 3D). VC_880_0 (max. *EF* = 7.0), VC_738_0 (max. *EF* = 7.8), and VC_771_0 (max. *EF* = 5.4) were strongly enriched in rain compared to foam and/or SML but were unrelated to viruses from public databases. Some VCs were overlap clusters defined as genomes sharing genetic overlap with other genome(s) belonging to multiple VCs. Enriched in rain, overlap cluster VC_634/VC_747 and overlap cluster VC_746/VC_747 were both related to *Pelagibacter* phage HTVC023P (max. *EF* = 7.0, Supplementary data 4), whereas overlap cluster VC_773/VC_829/VC_885 was related to *Flavobacterium* phage vB_FspM_immuto_3-5A (max. *EF* = 2.6). VConTACT2 detected various singletons and outliers, which usually represent new viruses, and had an *EF* > 8 for rain over sea surface ecosystems but were unrelated to any known virus. In boundary layer aerosols, e.g., VC_970_0 (max. *EF* = 1.2), VC_914_0 (max. *EF* = 2.1), and VC_738_0 (max. *EF* = 1.9) showed slight enrichments compared to marine samples but were also unrelated to any known viruses (Fig. 3d). Three singletons and nine outliers were additionally enriched in aerosols with outliers (*EF* = 2.8 and 1.4) being associated with a virus related to *Methylophilales* phage Melnitz−1 EXVC043M and *Vibrio* phage vB_VorS-PVo5, respectively. Overall, assembled marine viruses shared protein clusters with *Synechococcus*, *Rhizobium*, *Cellulophaga*, *Flavobacteria*, *Vibrio*, and *Pelagibacter* phages in vConTACT2[63] (Supplementary data 4, Fig. 5). Most positive correlations across all viromes were found between foam, SML, SSW, which are well-interconnected systems, and some positive correlations of specific marine samples with aerosol and rain samples were detected (Supplementary Fig. 6b, Supplementary data 14). Viromes of marine samples and rain samples were sometimes even negatively correlated, suggesting alternative sources of rain viruses other than the sea surface of the examined region.

We then investigated the aerosolization patterns of two circular viral genomes of similar lengths and carrying viral hallmark genes (terminases, portal protein) across the different ecosystems and stations (Fig. 6a, b). Virus_1 (39.7 kb, percent G/C base content = 46.7%, no RefSeq match in vConTACT2) was constantly of lower abundance in seawater samples compared to Virus_2 (35.1 kb, percent G/C base content = 35.1%, no RefSeq match in vConTACT2) across different stations. However, Virus_1 was consistently abundant in sea foams and was additionally found in three boundary layer aerosol samples and two rain samples. Instead, Virus_2 was absent from the boundary layer and rain samples despite its abundance in surface water. Virus_1 was linked to a *Porticoccus* MAG based on *k*-mer patterns (Supplementary Fig. 7) and single-nucleotide polymorphism (SNP) analysis revealed multiple SNP overlaps for this virus between a foam, aerosol, and rain sample, supporting its transfer from the sea surface to the boundary layer including rainwater sampled therein (Fig. 6c, d, Supplementary data 5). Mapping of reads from all samples against all 1813 viral scaffolds revealed shared viral populations between ecosystems (Fig. 7a). Sea foams, SML and SSW shared 837 viruses, whereas 15 viruses were present in all studied ecosystems. Overlaps between boundary layer aerosols and rain samples must be treated with caution, because small amounts of rain could have reached the aerosol filter membrane during sampling and filter exchange (Supplementary data 6), although we tried to rule out the second possibility by subtracting reads from handling controls. Precipitation had the highest number of viruses only detected in rain in this study (109), followed by foams (25), SSW (18), SML (7), and aerosols (6). Being exclusively detected in rain means that no other ecosystem from this study had 90% identical reads with 75% scaffold coverage of at least 1x for that virus. A percentage of 6.2% (112) of all viruses was shared between precipitation and seawater including foam. Interestingly, the rain sample pooled from February 14th to

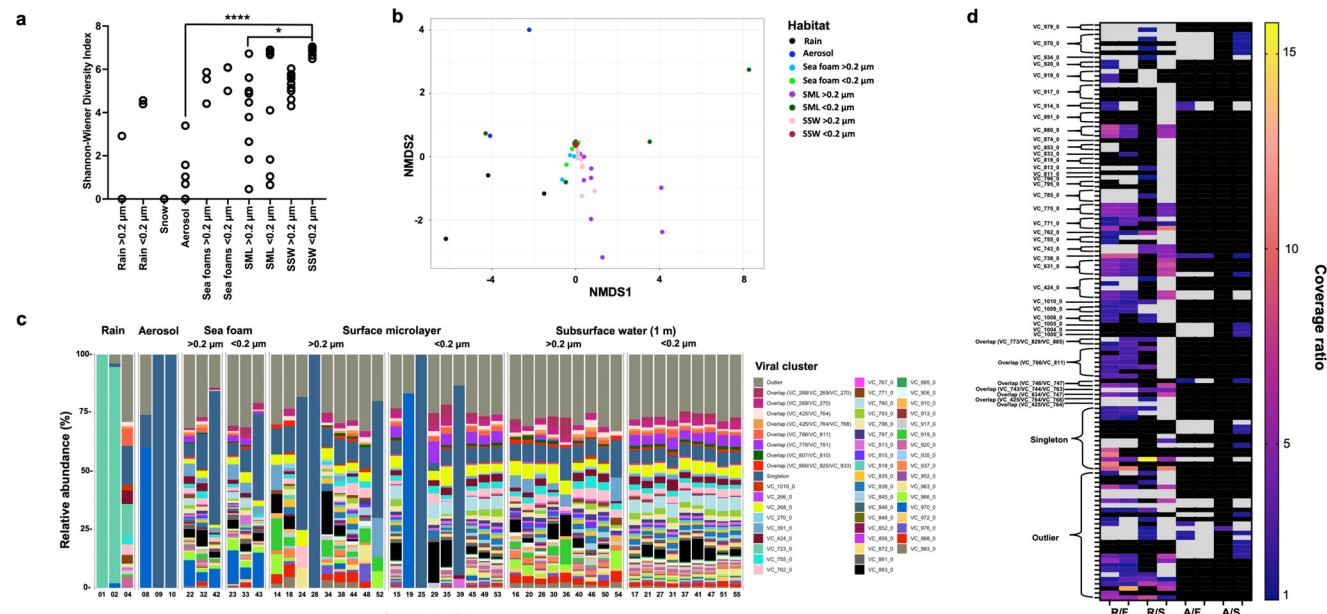

**Fig. 4 | Diversity and enrichment analysis of marine and airborne viruses based on relative abundance in rain, snow, aerosols, sea foam, surface microlayer (SML), and subsurface water (SSW).** Alpha-diversity for all samples (except for #12, which contains 0 viruses) depicted by Shannon-Wiener index; * = adjusted $p < 0.05$ (exact adjusted $p = 0.0293$), **** = adjusted $p < 0.0001$ in Dunn's multiple comparison test (post hoc analysis after Kruskal-Wallis test) for rain >0.2 μm ($n = 2$), rain <0.2 μm ($n = 2$), snow ($n = 1$), aerosol ($n = 7$), sea foam (both $n = 3$), SML (both $n = 9$), and SSW (both $n = 9$) samples (**a**), non-metric multidimensional scaling (NMDS) plot based on Bray-Curtis dissimilarity (stress = 0.06) (**b**), and stacked bar chart on beta-diversity (**c**). If samples only contained rare viruses (sample #7), a single virus (#3, #5, #6, #11 #13) or no viruses (#12), they were removed, and only the relative abundance of the 200 most abundant viruses assigned to viral clusters (VCs), outliers, and singletons were considered for (**b** and **c**). In vConTACT2,

outliers and singletons are unclustered and typically represent new viruses. Overlap clusters refer to genomes sharing genetic overlap with other genome(s) belonging to multiple VCs. In (**c**), marine samples are separated by size fraction: >0.2 μm = prokaryote fraction, <0.2 μm = viromes; Rain sample #1 is a > 0.2 μm sample, whereas #2 and #4 are rain viromes. Enrichment ratio of SML and foam viruses in rain and aerosols (**d**). Shown are ratios ≥1 of virus coverage for rain/foam (R/F), rain/SML (R/S), aerosol/foam (A/F), aerosol/SML (A/S), where the left tick stands for foam and SML 0.2 μm fraction and the right tick for foam and SML virome fraction in the denominator. Black fields mean that the virus was absent in one or both ecosystems in the respective sample. Grey areas show out of range fields (ratio between 0 and 1, indicating depletion). Sample number is explained in Supplementary data 12. Source data are provided as a Source Data file.

22nd 2020 (Event 2) defined most of this viral overlap compared to a sample from February 7th to 9th (Event 1). Based on read-mapping, Events 1 and 2 were associated with 22 versus 85 viruses assembled from marine samples as well as 112 versus 44 viruses assembled from rain, respectively. Event 1 was associated with 38 marine prokaryotic MAGs (min. 90% genome covered with reads), whereas 79 marine MAGs were found in the rain sample belonging to Event 2 (Fig. 8). To explain these differences by tracking to potential sources, backward trajectories (TJs) for air masses were calculated. They showed that during Event 2, air masses spent, during the first four days before arriving at the site, on average 72% of their time over the sea and loading conditions (loading of air masses with generic marine particles) were fulfilled on average 35% of the TJ. On the other hand, for Event 1, air masses spent less time above the sea (64%) and loading conditions were fulfilled, on average, only 10% of the TJ points (Fig. 8).

### Rain and aerosol viruses show adaptations toward their ecosystems and are targeted by marine prokaryote adaptive immunity

To investigate if rain and aerosol viruses have genetic adaptations, we explored the content of guanine (G) and cytosine (C) bases in viral scaffolds. Viral scaffolds solely detected in rain samples in this study ($n = 109$) exhibited a significantly higher percent G/C base content than total viruses found in rain (Kruskal-Wallis with Dunn's multiple comparisons test, $p = 0.0002$). All viruses found in marine samples had a significantly lower percent G/C base content compared to aerosol, total rain, and only rain viruses as detected in this study when compared pairwise (KW-test, $p < 0.0001$, Fig. 7b).

One very abundant circular viral genome was only detected in rain in the present investigation (VC_723_0, 39 kb, coverage = 189×, percent G/C base content = 59.7%, unknown family) with the closest relative of the VC being the *Rhizobium* phage RHph_N3_2. This phage carried typical phage hallmark genes like a major capsid protein, an endonuclease, a terminase and modification methylase, but also carried many hypothetical proteins (Supplementary data 7). In addition, two large viral scaffolds only detected in rain (270 kb and 496 kb) were *k*-mer linked to a *Flavobacteriaceae* MAG, encoded for sensors of blue-light using FAD (BLUF, Pfam/InterPro entry ID PF04940), a photoreceptor and for an UV-endonuclease *UvdE* (PF03851). A 16 kb-long viral genome with typical phage proteins (terminase, capsid) encoded for Tellurium resistance genes *TerD* (PF02342). From metagenomic assemblies, and from one MAG of *Schleiferiaceae* bacterium MAG-54, CRISPR arrays with evidence level 3 and 4 from 18 different samples could be detected by CRISPRCasFinder[64], and mostly belonged to marine ecosystems ($n = 14$) and rainwater ($n = 4$, Supplementary data 8). CRISPR spacers extracted across all samples based on consensus direct repeat (DR) sequences from recovered arrays matched protospacers of viruses from seawater, but also the viruses only detected in rain in this study (Fig. 5c, Supplementary Fig. 8). CRISPR spacers matching most viral protospacers were extracted from two dominant arrays, with one of them targeting primarily rain-derived viruses and the other one marine viruses (Supplementary Fig. 8, Supplementary data 8).

### Discussion

By detection of marine MAGs and viral genomes in marine, aerosol, and rain samples, our study showed that aerosolization from the sea

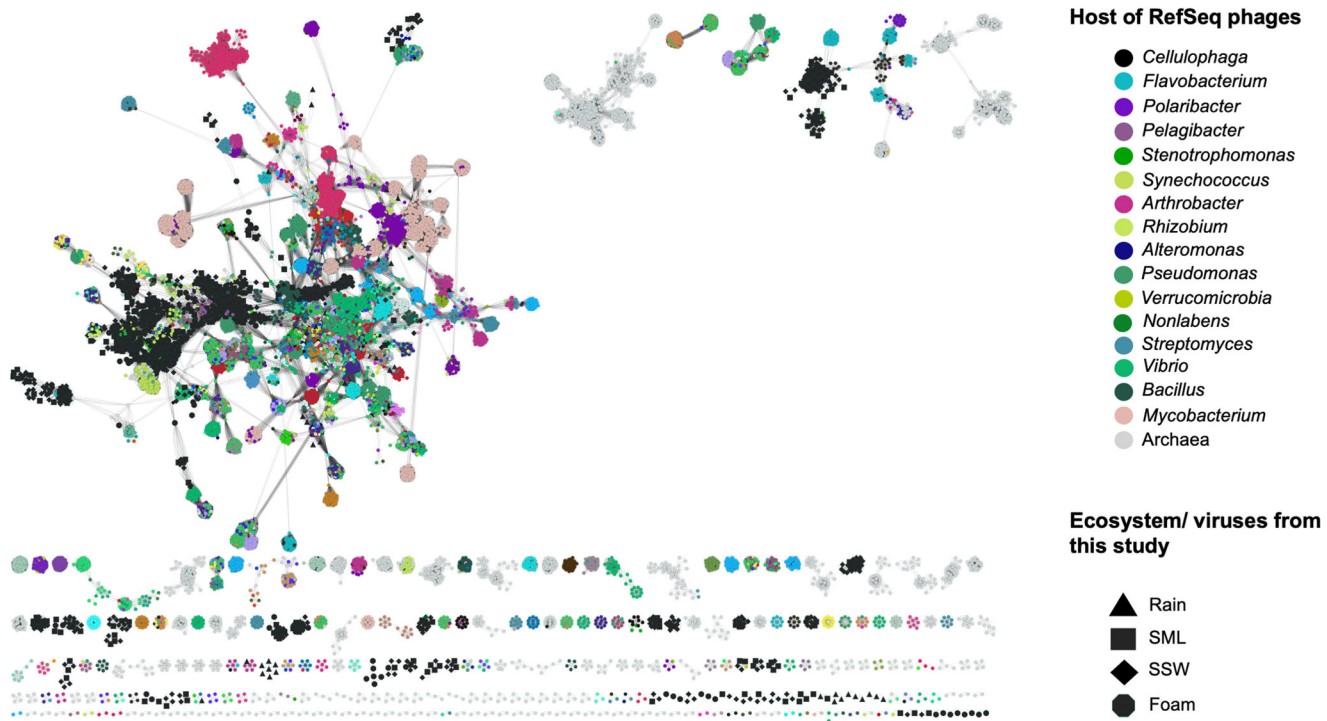

**Fig. 5 | Clustering of viruses from different ecosystems with viral genomes from the Refseq database (December 2021) reveals many clusters with unrelatedness to Refseq database viruses.** The legend has been reduced to hosts that show interactions with viruses from this study. Source data are provided as a Source Data file.

surface generally took place. While a previous mesocosm experiment showed that viral and bacterial aerosolization occurs taxon-specifically[35], our data now support that this process indeed happens in natural ecosystems. Viral attachment to biotic and abiotic surfaces in seawater is common[65] hence more likely occurring in the SML[18] and in foams (Fig. 1e). Since particles larger than 50 μm were removed prior to flow cytometry measurements, VLP counts from particle-rich foams were probably underestimated. On the other hand, methods based on fluorescence dyes are prone to generate fake VLPs[66], which could lead to the counting of false positives.

Rain samples shared 6.2% of the virome with marine samples indicating a notable viral exchange between both ecosystems, being supported by CRISPR spacers from sea surface prokaryotes matching viruses found exclusively in rain samples. Such established adaptive immunity indicates previous virus-host encounters along with viruses from rain leaving their signatures in the form of host-acquired CRISPR spacers in the sea surface and suggests that viruses are probably still infectious, i.e., can inject their genome into the host after deposition. Recent work has shown that highly populated ecosystems such as hydrothermal mats allow viruses to infect hosts across microbial domains[67], and that virus-host interactions can be specific to the SML within visible surface films[68]. At the same time protospacers can also be incorporated from defective phages[69], thus do not necessarily always indicate a successful replication of the virus. Our data revealed a tendency that at the air-sea interface, spacers from different CRISPR arrays targeted viruses from different sample origins (air, sea). We assume that these arrays have high turnover rates in this dynamic interface ecosystem, which will require more research. Atmospheric dispersal of viruses allows the spread of foreign genetic material into new habitats enabling bacterial evolution[70] and explaining the prevalence of similar viral genomes across large geographical distances[71]. Future work using culture-dependent experiments could elucidate if marine viruses not only remain infective after aerosolization[36], but also when deposited to Earth's surface with precipitation.

Rain and boundary layer aerosol viruses had a significantly higher percent G/C content compared to marine viruses. This feature had been attributed to carbon limitation[72], growth temperature[73], and avoidance of thymine-specific damage by UV radiation in bacteria[74] and occurred in recently described bacterial isolates from the stratosphere[75]. Since viruses and hosts were shown to have correlating G/C base contents even across kingdoms[76], we speculate that the here described viruses with high G/C base composition could infect hosts of similar nucleotide proportion. Validation via infection experiments with suitable virus-host systems isolated from atmospheric ecosystems will be needed to substantiate these assumptions. As genetic adaptations like the nucleic acid base composition will not change within hours, for instance shortly after aerosolization, we assume that viruses could be maintained in the boundary layer or atmosphere above for some time and further supplied by marine or terrestrial sources as shown for bacteria sampled over the major oceans[23, 77]. Alternatively, the viruses could have been derived from an unknown source (non-local, marine, or terrestrial) and could have been dispersed into the rainwater, or the rain scavenged biological material from the atmosphere on the way to Earth. Moreover, our data indicate that the air mass trajectory is crucial for understanding airborne microbial diversity and viral biogeography, being especially relevant for the highly influenced ocean-atmosphere interface[78]. We conclude that viruses disperse bidirectionally (from sea to air and vice versa) along the natural water cycle and by extended distribution that involves crossing interfaces and ecosystem boundaries they augment opportunities to shape microbial diversity and to contribute to biogeochemical cycles in their destination. Rainwater is a key component of the Earth's water cycle, and studying the microorganisms and viruses present in rainwater and their dissemination along the natural water cycle can help us better understand the cycling of water, pollutants, and nutrients in the environment. This can have further implications for water resource management, agriculture, and ecosystem health.

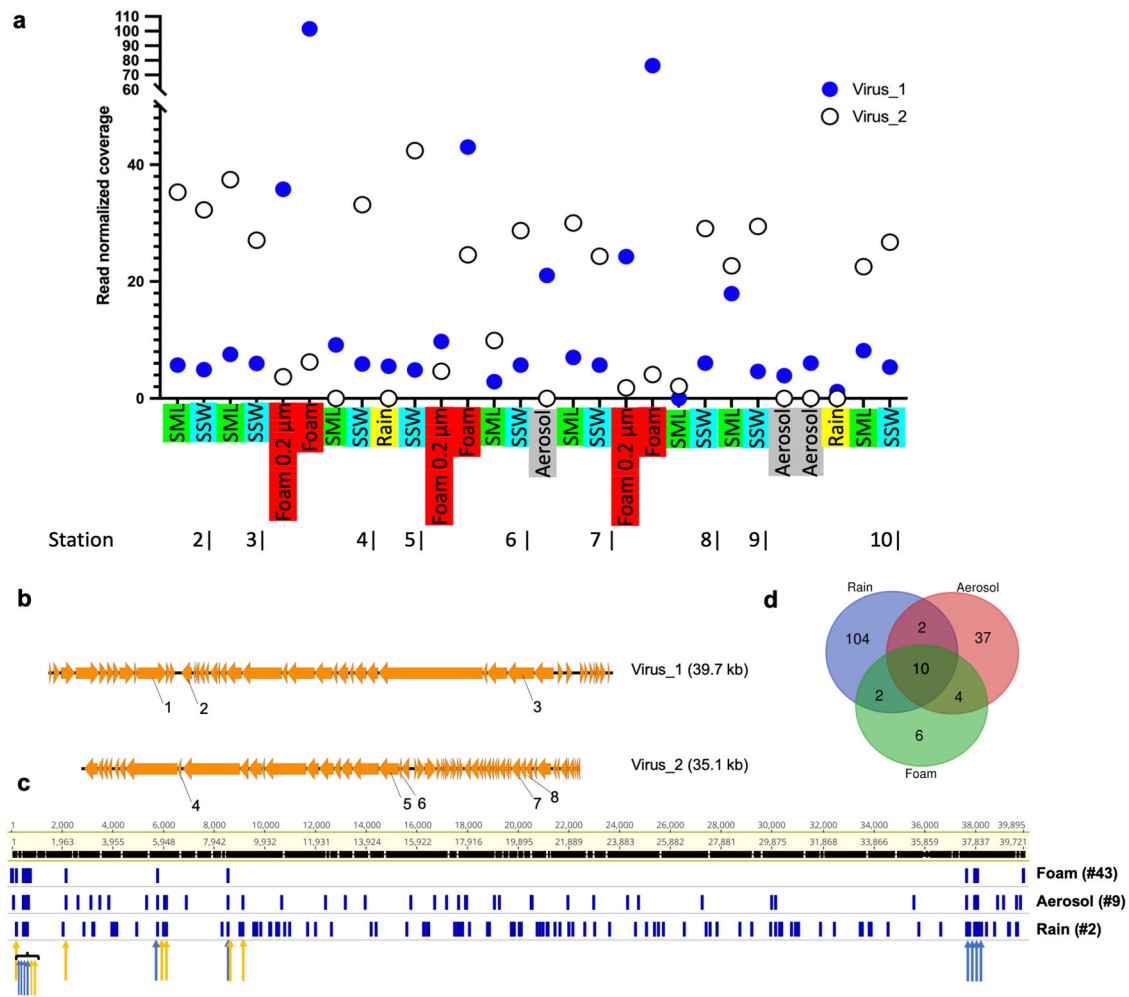

**Fig. 6 | Virus aerosolization and SNP analysis.** Succession of the coverage of two circular viral genomes across 28 metagenomes derived viromes of surface micro-layer (SML), 1-m deep subsurface water (SSW), and sea foam as well as from sea foam filtered onto 0.2 μm membranes, aerosol, and rainwater samples (**a**). Synteny and functional annotations of the two circular viruses visualized using Easyfig[82] and annotated with DRAM-v[80]. Functional annotations are 1: Bifunctional DNA primase/polymerase, N-terminal [PF09250.12], 2: Sec-independent protein translocase protein (*TatC*) [PF00902.19], 3: Terminase, 4: Concanavalin A-like lectin/glucanases superfamily [PF13385.7], 5: Phage P22-like portal protein [PF16510.6], 6: Terminase-like family [PF03237.16]; Terminase RNAseH like domain [PF17288.3], 7: C-5 cyto-sine-specific DNA methylase [PF00145.18], 8: PD-(D/E)XK endonuclease [PF11645.9] (**b**). Variant analysis of Virus_1 for a sea foam, aerosol and rain sample reveals overlapping nucleotide polymorphisms. Blue and orange arrows indicate overlaps between three and two samples, respectively. For details, please see Supplementary data 5 (**c**). Venn diagram showing variant overlaps for Virus_1 in different ecosystems as shown in c (**d**). Source data are provided as a Source Data file.

## Methods

### Seawater sampling and processing

Seawater sampling sites were located in the bay offshore Tjärnö, Swedish west coast in the Skagerrak (Fig. 1a), an area characterized by strong salinity gradients[79] (Supplementary data 9). Foams and SML were sampled from a small boat using the glass plate method[25,80]. In brief, a glass plate is immersed perpendicularly to the ocean surface into the water and withdrawn at a speed of 5–6 cm s$^{-1}$ [81]. Adhering surface film is scraped off from both sides of the plate with a squeegee blade into a collection bottle. Corresponding SSW from 1 m depth was collected as a reference using a syringe connected to a weighted hose. All equipment was treated with household bleach and pre-rinsed with sampling water. Wind speed was measured for ~1 min with a handheld VOLTCRAFT AN-10 anemometer (Conrad Electronic, Hirschau, Germany) held at 2–3 m above the sea surface and either an approximate average was reported, or a range in case of stronger variations (Supplementary data 9). Light conditions were recorded on the boat using the Galaxy Sensors smartphone application v.1.8.10. Temperature and salinity were measured at ~20 cm beneath the surface from the small boat using a portable thermosalinometer (WTW™ MultiLine™ 3420).

Water samples were stored in the dark and on ice until processing in the laboratory. Filtration equipment was treated prior to all usages with household bleach and rinsed with MilliQ water. Seawater (500 mL SML and 2 L SSW) and sea foams (200–400 mL) were sequentially vacuum filtered through 5 μm and 0.2 μm pore size Omnipore PTFE filter membranes (47 mm diameter, Merck/Sigma-Aldrich, Darmstadt, Germany). The flow-through of the 0.2 μm filter membrane was pre-cipitated with 1 mg L$^{-1}$ iron-III-chloride (Alfa Aesar/Thermo Fisher Scientific, Uppsala, Sweden) for 1 h at room temperature[82], and the flocculates were in turn filtered onto another 0.2 μm Omnipore PTFE filter membrane to obtain viruses and small prokaryotes. All filters were stored at −80 °C until further processing and shipped on dry ice to the home laboratory for DNA extraction from the 0.2 μm filter and the FeCl₃ flocculates.

### Aerosol and precipitation sampling

We used a land-based aerosol pump/constant flow sampler (QB1, Dadolab, Milan, Italy) with a custom-made filtration unit (SIMA-tec GmbH, Schwalmtal, Germany) to filter aerosols from the atmosphere in coastal proximity about ~2 m over ground between buildings

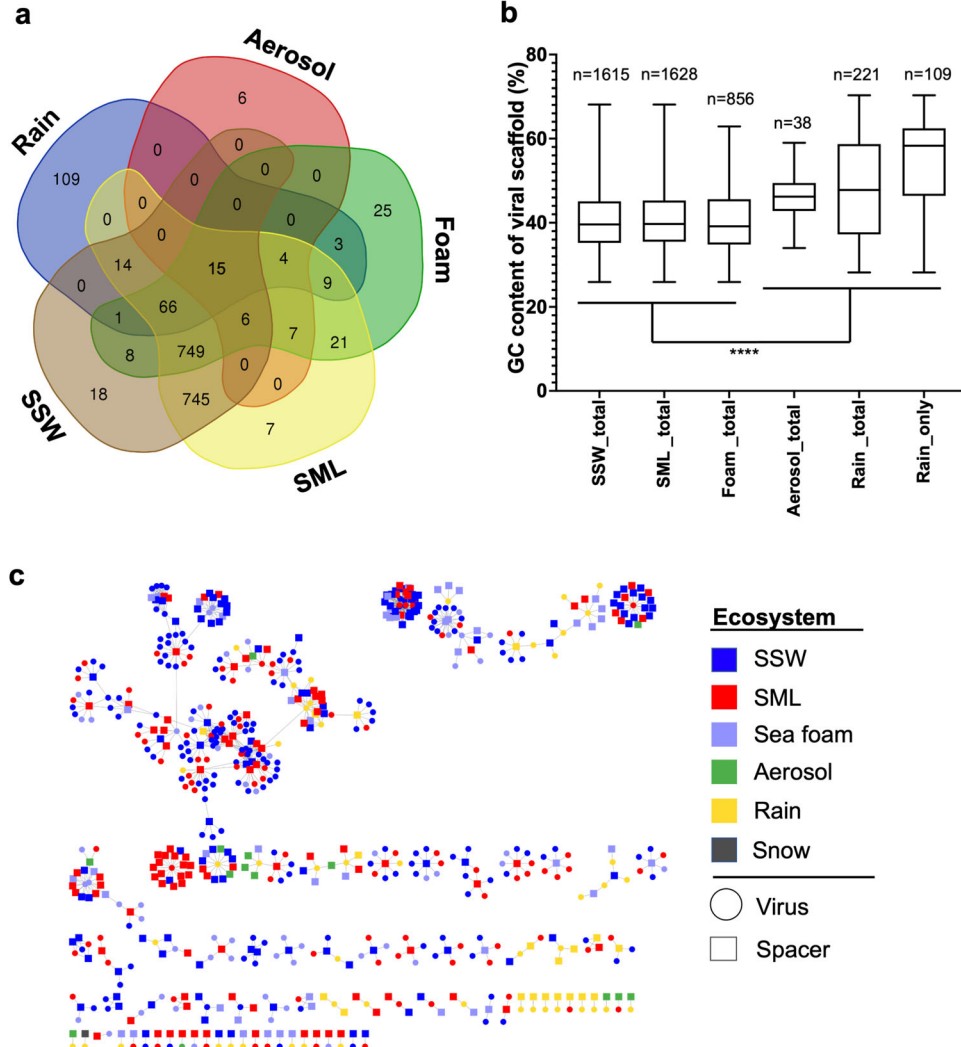

**Fig. 7 | Overlapping occurrence of viral scaffolds, their percent G/C base content and CRISPR spacer to viral protospacer hits.** Overview of shared viral scaffolds (>10 kb length) between seawater, aerosol, and precipitation obtained from 55 metagenomes and determined by the mapping of reads. A viral genome was considered present in a sample if at least 75% of the genome were covered with reads at least 90% identical to the genome, in accordance with suggested viromics benchmarks[78] (**a**). Percent of the bases guanine (G) and cytosine (C) in viral scaffolds from rain, aerosol, foam, surface microlayer (SML), and subsurface 1-m deep water (SSW) based on read mapping. Rain_only refers to viral genomes exclusively found in rain in this study. Stars indicate significant differences after Kruskal Wallis test and Dunn's multiple comparison test (****, adjusted $p$ = <0.0001). In each pairwise comparison, the marine groups were significantly different from the atmospheric groups. Rain_total was also significantly different from Rain_only (***, adjusted $p$ = 0.0002), which is not indicated to reduce complexity of the figure. The line of the box plot represents the median, the box extends from the 25th to 75th percentiles, whiskers indicate the min. to the max. value (**b**). CRISPR spacers (origin indicated as square) matching assembled viral scaffolds (circles) derived from different ecosystems (**c**). Source data are provided as a Source Data file.

(Supplementary Fig. 9). This height is not relevant for a characterization of atmospheric aerosols including cloud condensation nuclei and INP but does provide information on seaborne aerosols and their role as viral and microbial vehicles. Incoming air was filtered through 0.1 μm pore sized Omnipore PTFE filter membranes (Merck/Sigma-Aldrich). Filtered volumes and filtration duration varied and ranged from 19 to 61 m³ (average volume flow 7 L min⁻¹) and from 24 to 96.5 h, respectively (Supplementary data 6). The volume was normalized to the mean temperature and mean air pressure from the start and end of an aerosol filtration. Handling controls for aerosol samples were collected as follows: a filter membrane was briefly placed on the filter unit, and directly frozen in a falcon tube at −80 °C. Snow and rain with a volume of 90 mL and 150 to 1050 mL, respectively were collected using funnels taped to Duran glass bottles. Rain was collected and, like seawater, filtered onto 0.2 μm pore size PTFE filter membranes, and the viral fraction was obtained as explained above. Rain collected between 14th to 22nd of February 2020 was prefiltered onto 5 μm due to visible

pieces (probably plant-based) in the sample. To achieve sufficient DNA yield for sequencing, DNA from rain for the periods 07th to 09th of February and 14th to 22nd of February 2020 were pooled, respectively. The snow sample was prefiltered on 5 μm and frozen at −80 °C. Later in the home laboratory, it was thawed at room temperature and concentrated in an Amicon® Ultra-15 centrifugal filter unit (Ultracel 100 kDa, Merck Millipore, Darmstadt, Germany) by spinning in several steps at 3000 × $g$, 10 min. at 4 °C before DNA extraction. More details on rain and snow samples can be obtained from Supplementary data 10.

**Air mass paths (backward trajectories)**
Transport pathways of air masses were evaluated with 5-day backward trajectories (TJs) generated using the Hybrid Single-Particle Lagrangian Integrated Trajectories (HYSPLIT) model[83]. The TJs were calculated every one hour ending at 700 m above the site for the period 1st to 29th February 2020. The European Centre for Medium-range Weather

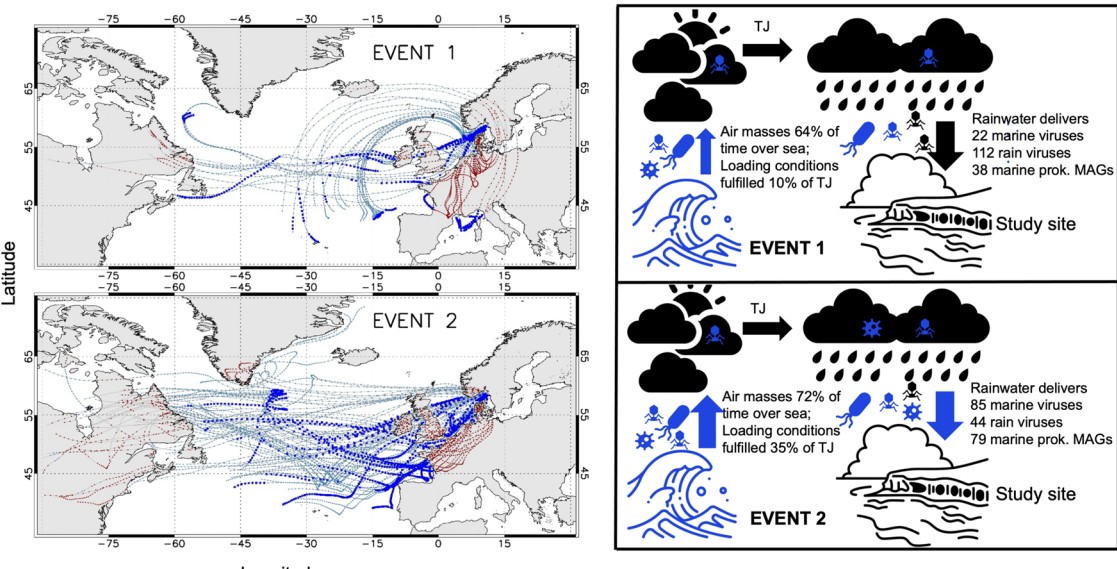

**Fig. 8 | Backward trajectories (TJs) of two rain events leading to different deliveries of marine viruses and metagenome-assembled genomes (MAGs) at the study site.** Event 1 (upper left panel) refers to a rainwater sample from 7th to 9th of February, and Event 2 (lower left panel) to a sample collected between 14th to 22nd of February 2020. Sky blue and red points highlight where TJs travel above sea and land, respectively. Blue filled squares represent points where loading conditions were fulfilled (please see main text for further explanation). Panels on the right show corresponding deliveries of viruses and MAGs to the study sites. A virus was considered marine or from rain if assembled in such a sample and counted if detected based on read mapping. Marine MAGs were considered present in rainwater 0.2 μm samples if 90% of the genome was covered with reads (see Supplementary data 3). The right panel figure was created using Adobe Express. Source data are provided as a Source Data file.

Forecasts (ECMWF) ERA5 model atmospheric reanalysis[84] was used to initialize HYSPLIT. After five days, the uncertainty associated with TJs is estimated between 10 and 30% of the travel distance[85]. Each TJ was then projected on the 10-m wind, total precipitation, land mask, surface pressure and cloud fraction model fields (ERA5), associating each point along the path with the nearest values of the considered model variables. The choice of the ending height (700 m) above the site is based on the analysis of in situ meteorological (Nordkoster A Automatic Weather Station, 58.890 °N, 11.010 °E as obtained from SMHI, https://www.smhi.se/data/meteorologi/ladda-ner-meteorologiska-observationer/#param=lowestCloudBaseInstant,stations=all,stationid=81540) and model (ERA5) data for February 2020 (Supplementary Fig. 10). To identify loading areas and air masses presumably responsible for the transport towards the site, a selection of TJs was carried out considering those (ones) arriving above the site during precipitation sampling Events 1 (7–9 February 2020) and 2 (14–16 and 20–22 February 2020). Event 1 TJs were associated with an extratropical cyclone (Storm Ciara), which mainly affected the United Kingdom, but also crossed northern Europe[86]. Similar to Becagli, et al.[87], loading conditions along TJs were evaluated searching where each TJ was within the mixing layer and wind speed at surface was greater than 3 m s[-1]. TJ analysis was performed using Interactive Data Language (IDL) software v. 8.7.2.

### Microbial cell counts and virus-like particle abundances

Duplicates of unfiltered seawater, foam, and precipitation samples were fixed with glutardialdehyde (1% final concentration, Merck, Sweden), stored for 1 h in the dark and subsequently stored at −80 °C. Particle-enriched foams were gravity filtered onto 50 μm filters (Cell-Trics®, Sysmex Partec, Muenster, Germany) before cell counts of prokaryotes and small phototrophic (autofluorescent) eukaryotes were measured by a BD Accuri C6 flow cytometer (Becton Dickinson Biosciences, Franklin Lakes, USA) according to established protocols[88–90]. Prokaryotic cell numbers were determined after the protocol of Giebel, et al.[90]: In brief, the in an ice-bath thawed sample was stained by the DNA dye SYBR® Green I (10x final concentration,

Invitrogen/Thermo Fisher Scientific, Carlsbad, CA, USA). As internal standard and for performance monitoring, 1 μm multifluorescent latex beads (Polysciences Europe, Eppelheim, Germany) were used. After 30 min. of incubation in the dark, each sample was analyzed for 2 min. using a flow rate of 14 μL min.[-1]. Samples with an event rate >1500 events s[-1] were diluted with sterile seawater to avoid coincidence. Small eukaryotic phototrophic cell numbers were determined after the protocol of Giebel, et al.[89] and Marie, et al.[88]: Slowly thawed (ice bath) and unstained sample was mixed with internal standard beads and subsequently analyzed for 3 to 4 min. using a flow rate of 66 μL min[-1]. Due to previously reported low coefficient of variance among SML biological replicates in flow cytometry[91], we did not measure biological replicates. VLPs were determined following exactly the protocol of Brussaard, et al.[92] using samples fixed with glutardialdehyde (final concentration 1%). In brief, samples were diluted with a 0.02 μm filtered TE buffer (10 mM Tris, 1 mM EDTA, pH 8.0, Merck/Sigma-Aldrich) and stained with a final concentration of 0.5% SYBR Green I (Invitrogen/Thermo Fisher Scientific, Carlsbad, CA, USA) for 10 min. at 80 °C and a 5 min. cooling period. For VLP counts, the event rate was kept below 1000 events s[-1]. The gating strategy is shown in Supplementary Fig. 11 and was implemented in the BD Accuri Flow C Software v.1.0.0264.21, build 20120423.264.21. Flow cytometry results were further compared to VLPs counted under the epifluorescence microscope (see below). Enrichment factors (*EFs*) as a standard parameter in the research field were calculated by taking the ratio of a specimen (e.g. number of cells or VLPs) in the SML to its SSW counterpart. Calculating *EFs* does not consider that the residence time of cells and VLPs in SML is unclear. *EF* >1 and <1 indicate an enrichment and a depletion of measured specimens, respectively.

### Epifluorescence microscopy

Representative samples covering all ecosystem types and the abundance range of the whole sample set were additionally counted using epifluorescence microscopy to validate VLP numbers based on flow cytometry. Filters for virus quantification were prepared following the

standard protocol by Suttle and Fuhrman[93]. In brief, samples were diluted using 0.02 μm-filtered phosphate-buffered saline (VWR, Darmstadt, Germany), filtered onto 0.02 μm Anodisc filters (Whatman, Maidstone, UK) by applying vacuum, stained with SYBR Green I (20 x concentration, Thermo Fisher Scientific, Waltham, MA, USA) for 15 min. in the dark and mounted onto microscopic slides with 0.1% p-phenylenediamine (Acros Organics, Geel, Belgium) as antifade solution. A minimum of 300 VLPs were counted per filter in at least 15 randomly chosen counting grids at a 1000× magnification on a Leica DMRBE Trinocular (Leica Microsystems, Wetzlar, Germany) using the software EOS Utility v.2.10.2.0. Analysis of microscopic images was performed in ImageJ v.1.5.2[94].

## Ice-nucleating particles
INP was measured from the 5 μm filter membrane that was used for the pre-filtration of seawater samples. Of these filters, small disks with 1 mm diameter were punched out, using biopsy punches, and each disk was immersed in 50 μL of ultrapure water in a well of a 96-well PCR tray (BrandtTech®, Essex, CT, USA). For each filter membrane, 24 punches were examined, filling one-quarter of a PCR-tray. The PCR-tray was then sealed and cooled down in an ethanol bath of a thermostat with a cooling rate of 1 K min.$^{-1}$, while a camera took pictures every 0.1 K from above. In these pictures, frozen wells can be well distinguished from unfrozen ones, and the cumulative number of frozen wells was assessed for the different samples, a clean filter and pure water. Concentrations of INP were calculated from the cumulative number of frozen droplets, based on the known amount of filtered water and Poisson statistics[95,96]. Data were plotted using OriginPro 2020 (64-bit) SR1 9.7.0.188 (Government).

## Statistical analyses
Correlations between abundances of prokaryotic cells, small phototrophic eukaryotes and VLPs were investigated using the cor.test function in R version 4.0.3.[97] within R studio v.1.3.1093[98]. Pearson correlations were applied after the Shapiro-Wilk test confirmed normal distribution of data and residuals (for linear models), otherwise Spearman rank correlation was chosen. Dependences of *EFs* on environmental variables (wind speed, light, salinity) and interactive effects of those parameters were further investigated using linear regressions, and the models were validated using adjusted $R^2$ and *AIC* in the R programming environment. Differences in alpha-diversity and viral percent G/C base content were analyzed using a Kruskal-Wallis test with Dunn's multiple comparison as post hoc analysis in Graphpad Prism v.9.4.1. Ecosystem-based differences in beta diversity shown in NMDS plots were assessed using Permutational multivariate analysis of variance (PERMANOVA, $n = 999$ permutations) as well as Betadispersion analyses followed by a TukeyHSD test and executed by 'adonis2' and 'betadisper' function of the R package vegan v.2.5-7[99], respectively.

## DNA extraction and sequencing of metagenomes
Genomic DNA was extracted from seawater (0.2 μm and <0.2 μm flocculated viral fraction), rain filter membranes (47 mm diameter, Merck/Sigma-Aldrich), and the concentrated snow sample using the DNeasy PowerSoil Pro Kit (Qiagen, Hilden, Germany). DNA from aerosol filters (90 mm diameter, Merck/Sigma-Aldrich) was extracted using DNeasy PowerMax Soil Kit (Qiagen) with a subsequent DNA precipitation step. After concentration in a speed-vac Concentrator plus (Eppendorf AG, Hamburg, Germany), DNA was quantified using Qubit™ dsDNA High Sensitivity Assay Kit on a Qubit™ 4 Fluorometer (Invitrogen/Thermo Fisher Scientific) and sent for metagenomic sequencing to Fulgent Genetics (CA, USA). Library preparation was done according to the Illumina DNA Prep with Enrichment Reference Guide (Document # 1000000048041 v05, June 2020). FastQC[100] did not detect any elevated sequence duplication levels or over-represented sequences.

## Metagenomic analyses
Raw shotgun sequencing reads of seawater (foams, SML, SSW), aerosols, and precipitation datasets were quality-trimmed using bbduk (https://github.com/BioInfoTools/BBMap/blob/master/sh/bbduk.sh)[101] and Sickle v.1.33[102].

Sequencing controls were assembled using MetaSPAdes v.3.13[103] and used as a blueprint for read mapping[104] of actual samples; any reads that mapped to the negative controls were removed from downstream analyses (https://github.com/ProbstLab/viromics/blob/master/extract_unmapped_stringent/extract_unmapped_stringent.sh). The same procedure including handling controls was carried out for metagenomic reads of aerosol samples.

Within a Snakemake[105] workflow designed for detecting viruses and prokaryotes, quality-controlled paired-end reads were first assembled with MetaviralSPAdes v.3.14.0[106] and reads were mapped back[104] to the assembly. Unassembled reads were assembled using MetaSPAdes v.3.14[103], and the two assemblies were joined for downstream processing. VIBRANT v.1.2.1.[107], VirSorter v.1[108] (only category 1, 2, 4, 5 were considered) and ViralVerify v.1.0[106] were used to identify viral scaffolds and host contamination was removed with CheckV v.0.7.0 (database v.0.6)[109]. Only viral scaffolds >10 kb were kept and clustered at the species level (95% similarity) using VIRIDIC v.1.0 r3.6[110], and the longest or circular scaffold of each cluster was used as representative. Metagenomic reads were mapped to >10 kb viral genomes with at least 90 % identity using Bowtie2 v.2.3.5.1[104] with settings --ignore-quals −mp = 1,1 −np = 1 −rdg = 0,1 −rfg = 0,1 --score-min = L,0,-0.1[111]. To show the succession of two circular viral genomes across different samples, a separate mapping was done for these two scaffolds, and SNP analysis was performed for one marine virus that got airborne using Geneious v.11.1.5[112] with default settings for variant analysis. Venn diagrams were constructed using ugent Webtool (https://bioinformatics.psb.ugent.be/webtools/Venn/). Complying to current viromics conventions[113], only scaffolds covered with 75% of reads were considered further, and breadth was checked with calcopo (https://github.com/ProbstLab/viromics/tree/master/calcopo/calcopo.rb)[114]. Mean coverage of viral scaffolds was calculated (https://github.com/ProbstLab/uBin-helperscripts/blob/master/bin/04_01calc_coverage_v3.rb)[61], and sum-normalized based on sequencing depth. Genes on viral scaffolds were predicted using Prodigal v.2.6.3[115] in meta mode and functionally annotated using DRAM-v v.1.2.4[116]. Synteny of viral genomes was visualized using Easyfig v.2.2.5[117]. Clustering of dereplicated viral genomes with a RefSeq database (release Dec. 2021, taken from https://github.com/RyanCook94/inphared)[118] was performed using vConTACT2 v.0.9.19.[63,119]. Information on VCs and closest relative were compiled using graphanalyzer v.1.5.1 (https://github.com/lazzarigioele/graphanalyzer)[120], and networks visualized in Cytoscape v.3.9[121]. Further viral taxonomic information was inferred from PhaGCN2.0[122,123]. Relative abundance of VCs was used for beta-diversity analysis. The % G/C base content of viral scaffolds counting towards a sample if detected based on read mapping, was calculated with an inhouse script (https://github.com/ProbstLab/uBin-helperscripts/blob/master/bin/04_02gc_count.rb)[61]. For investigating aerosolization and enrichment of VCs in rain over SML and foam, the maximum (sum-normalized) coverage of a virus across an ecosystem, e.g., across all aerosol samples, was considered assuming this value represents the highest possible abundance in that ecosystem. Then coverage ratios were calculated for the pairing rain/foam (R/F), rain/SML (R/S), aerosol/foam (A/F), aerosol/SML (A/S), and foam and SML samples were distinguished between the 5−0.2 μm and virome fraction.

## Prokaryotic community composition, binning of MAGs, and virus-host interactions
Genes from combined scaffolds from the two assembly steps were predicted using Prodigal v.2.6.3 in meta mode[115], and genes were

annotated using DIAMOND v.0.88[124] blast against FunTaxDB v1.1 (https://zenodo.org/record/7180192#.Y6Mjn-LMLtN)[61]. Ribosomal protein S3 (rpS3) genes were retrieved via word searches from the annotations (excluding type a/e) and clustered at 99% identity using CD-HIT v.4.8.1[125], and the scaffold of the centroid of the cluster was used for downstream mapping (Bowtie2) of individual samples. Taxonomic assignment of rpS3 genes was performed using USEARCH v.10.0.240_i86linux64[126] against the rpS3 taxonomy database by Hug, et al[127]. Any unclassified and any eukaryotic hits were excluded and coverages were read-sum normalized. For mean relative abundances taxonomic units of the same taxonomy were summed up. Analysis of the Shannon-Wiener Index (alpha-diversity) using the estimate_richness function, beta-diversity, and NMDS analysis based on Bray-Curtis dissimilarity with the ordinate function were performed using the phyloseq package v.1.34.0[128] in R version v.4.0.3.[97] within R studio v.1.3.1093. NMDS and beta-diversity plots were made using ggplot2 v.3.3.5.[129]. Binning of MAGs was done using MetaBAT v.2.15[130] and Maxbin2 v.2.2.7[131] with aggregating best genomes by DasTool v.1.1.1[132] and followed by manual curation in uBin v.0.9.14.[61]. MAGs were quality-checked using CheckM v.1.1.3[62], and taxonomic assignment was performed with the classify workflow of GTDB-tk v.1.7.0 (database release r202)[133]. Mapping to individual MAGs was performed with Bowtie2 under allowance of 2% error rate (3 mismatches) for breadth calculation. Virus-host interactions were inferred from CRISPR-spacer matches and shared k-mer frequency patterns between assembled viruses and host MAGs. At first, CRISPRCasFinder v.4.2.20[64] with -minDR 16 was run on sample assemblies >1 kb and MAGs to find CRISPR consensus DR sequences from arrays with ≥ evidence level 3. Consensus DR sequences with 100% similarity hits to viral scaffolds were removed from further analysis. Then DRs were used in MetaCRAST[134] with settings -d 3 -l 60 -c 0.99 -a 0.99 -r to extract CRISPR spacer from the read files of each sample. Spacers were homopolymer and length-filtered (20–60 bp), clustered at 99% identity, BLAST was performed with a BLASTn –short algorithm[135] against the viral scaffolds, and filtered at 80% nucleotide similarity. Prokaryotic MAGs (116) were compared and dereplicated using dRep v.3.2.2[136] at 95% average nucleotide identity. Additional MAGs that were excluded in the dRep process due to low quality in CheckM but had good contamination/completeness scores in uBin and formed their own cluster in the dRep compare mode were additionally considered for k-mer based virus-host linkages. Viral scaffolds were assigned to these MAGs using VirusHostMatcher v.1.0.0[137] at a d2* threshold of 0.3, as previously performed[138]. Spacer-protospacer interactions and virus-host interactions based on k-mers were visualized using Cytoscape v.3.9[121].

## Reporting summary
Further information on research design is available in the Nature Portfolio Reporting Summary linked to this article.

## Data availability
ECMWF ERA5 reanalysis data are freely available on Copernicus Climate Change Service (C3S) Climate Data Store (CDS) under: https://doi.org/10.24381/cds.bd0915c6 and https://doi.org/10.24381/cds.adbb2d47. Flow cytometry data have been stored at PANGAEA database[139] and linked to the Integrated Marine Information System (IMIS). Epifluorescence microscopy pictures[140], trajectories[141], and the viral network[142] are available at figshare. The Viral Refseq Database (release December 2021) is available from INPHARED[118]. All sequencing data, MAGs, and the viral metagenome are stored in Bioproject PRJNA811790. For further details on accession numbers, please refer to Supplementary data 11. Source data are provided with this paper.

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

## Acknowledgements

We acknowledge funding for the MIDSEAS (Microbial dispersal from air to sea and snow) project by ASSEMBLE PLUS (European Union's Horizon 2020 research and innovation program, Grant Agreement No. 730984), and by the German Aerospace Center (DLR) for the project DISPERS (50WB1922). JR received funding by the German Research Foundation (DFG RA3432/1-1, project number 446702140 and WBP Return Grant RA3432/1-3, project number 534276621). JP was supported by Aker BP within the framework of the GeneOil Project. MEH received funding within the framework of the PhD research-training group "The Ecology of Molecules" (EcoMol) funded by German Research Foundation (DFG) within the TRR 51 Collaborative Research Center (CRC) "Roseobacter". HAG was funded as well by the DFG, within the CRC TRR 51, grant/award no. 34509606. AJP received funding by the Ministerium für Kultur und Wissenschaft des Landes Nordrhein-Westfalen ("Nachwuchsgruppe Dr. Alexander Probst") and the DFG (PR1603/2-1). Open Access funding was enabled by Projekt DEAL. We like to thank the Sven Lovén Centre for Marine Sciences, Tjärnö, Sweden as part of the University of Gothenburg for hosting JR and SPE during the campaign and providing excellent science support including the use of laboratory and boat facilities. Here, we especially like to thank Anna-Karin Ring, Kerstin Johannesson, and Joel White. We further thank Sabrina Eisfeld for laboratory maintenance, and Amelie Assenbaum for technical assistance during INP measurements. We also thank Ken Dreger for server administration, as well as Svenja Parge and Till Bornemann for bioinformatics assistance. We thank Christopher Pöhlker, Daniela Meloni, and Silvia Becagli for discussions about aerosol sampling. The data handling was partially enabled by resources provided by the Swedish National Infrastructure for Computing (SNIC) at UPPMAX partially funded by the Swedish Research Council through grant agreement no. 2018-05973. Pavlin Mitev at UPPMAX is acknowledged for assistance.

## Author contributions

J.R. conceptualized the study, conducted data analysis, wrote the first draft of the manuscript, and conducted field sampling together with S.P.E. J.P. and A.J.P. developed the viromics pipeline and together with S.P.E. provided bioinformatic assistance. H.A.G. measured and analyzed cell counts and together with M.E.H. determined VLP counts. M.E.H. conducted epifluorescence microscopy. H.W. provided data on ice-nucleating particles. A.S. conducted binning of prokaryotic MAGs. C.S.

and P.G. calculated backward trajectories. A.J.P. provided supervision, bioinformatic guidance, help with analysis, scripts, and resources. All authors contributed to the writing and editing of the final manuscript.

## Funding

## Competing interests
The authors declare no competing interests.
