## [Peer Review File · Nature Communications]

Marine viruses disperse bidirectionally along the natural water cycleEditorial Note: This manuscript has been previously reviewed at another journal that is not operating a transparent peer review scheme. This document only contains reviewer comments and rebuttal letters for versions considered at *Nature Communications*.

REVIEWER COMMENTS

Reviewer #3 (Remarks to the Author):

The authors addressed most of my comments, and I think the “flow” of the manuscript at its present form is much improved.

Still, I cannot agree with reply 32, and some clarifications are needed:

Why did the authors provide back trajectories only for the rain samples? Is there a logic behind not providing this information also for the air samples? As the author indicate in their conclusions (L. 378): “...the air mass trajectory is crucial for understanding airborne microbial diversity and viral biogeography, being especially relevant for the highly influenced ocean-atmosphere interface.”

If no special reason for the lack of this analysis, please provide air mass trajectories on the air samples as well.

Reviewer #4 (Remarks to the Author):

The manuscript “Heads in the clouds: Marine viruses disperse bidirectionally along the natural water cycle” by Rahlff et al., compared metagenomics analysis of measurements performed at 1m depth water, the SML, and foams off the shore in the bay of Tjärnö, Sweden, and aerosol and rain collected on a station located at the shore, and use these analyses to describe exchange between the ocean surface, aerosol particles and rain. While their metagenomic analysis is very well done and robust, I have many constraints on their conclusions and the methodology. After reading the revised manuscript and the answer to the comments of the previous reviewers, I cannot endorse publication of the article in the present form, the article needs major corrections. I hope my comments are helpful to the authors.

The way the article is framed underlies a misunderstanding of cloud and rain formation. This was highlighted to me in the answer to one of the reviewer's comments. A previous reviewer correctly requested, as a minor comment, to not use the term "unique to rain", and the authors responded by asking "We would be interested to know why the viruses should clearly originate from somewhere else (where from)? The viruses we report here were only detected in rain samples." This quite unfortunate answer reflects the lack of understanding of cloud and rain formation. Clouds in our atmosphere cannot form without aerosols, for water to change phase homogeneously it needs about 400% supersaturation values. These values are impossible to get in our atmosphere; therefore, a nucleolus/surface (i.e., aerosols) is needed for the water vapor to condense. Aerosols are either emitted from the surface (primary aerosols) or are formed directly in the atmosphere (secondary aerosols). Secondary aerosols are produced from the oxidation of volatile/semi-volatile organic compounds, and when they are formed they produce aerosol sizes on the order of 20nm or slightly bigger. Given that viruses do not form by oxidation and are much larger in size, they have to originate as primary aerosols, from what part of the planet and whether they are attached to another particle (e.g., bacteria) or "free-living" is a very interesting question.

Also, the authors mentioned in their introduction that viruses (and bacteria) were already found in clouds and that they might trigger their own precipitation, so we know viruses exist in clouds and precipitation, but we don't know which ones, we don't know if they can "live" in the atmosphere, we don't know how far they can be dispersed, we don't know if they can adapt to atmospheric conditions. These are questions the authors can advance our knowledge in.

Another constraint I have is their aerosol measurements; they are not atmospherically relevant. Aerosols collected at 2-m above the surface next to a hut are irrelevant to the boundary layer and even less to cloud processes. Those aerosol measurements are only relevant for microorganism diversity to their immediate surroundings. To be able to relate aerosols that might reach the cloud condensation level they need to be taken at least 10m above the surface. So in the context of the message of this article, those measurements cannot be used.

Why did the authors neglect two of the four rain events into their discussion? Given that half of your measurements did not contain enough DNA to sequence is a result in itself. How much rain water was collected in each rain event? In Fig. S12 it looks that one of the rain events (9-10 feb) that did not have enough DNA was larger than one (7-9 feb) that did have enough DNA. By the way, the figure is difficult to read.

In the first sentence of the Discussion, the authors state that by detecting marine MAGs and viral genomes in aerosol and rain samples, their study shows that aerolization from the sea surface took place. Why? The rain does not come from deep convective clouds that formed around the measuring site, most likely the rain comes from stratocumulus decks travelling several 10s if not hundreds of kilometers. Moreover, there is no mention of wind speeds and direction of when the SML, foam, and SSW measurements took place, so it is difficult to assess if some viruses were emitted before a rain event and maybe scavenged into the samples, or if the emitted particles from the sea surface stations could be transported in the direction of the aerosols and rain samplers. A previous reviewer even offered an opposite scenario (i.e., deposition of cloud-born –or the scavenged below-cloud airborne microbes), and again, instead of answering a very helpful and insightful comment to make the manuscript better, the authors answer with a question. How do the authors know they aerosols were not scavenged? “highlighting” in the abstract the “bidirectional route” does not answer the comment from the previous reviewer. Furthermore, bidirectional should not be used, it is a misleading word that assumes the viruses originated from the atmosphere, and as I explained in my first comment this is not physically possible.

One of the reviewers also mentioned the lack of ecological context of the manuscript, and that the text is very technical. The revised manuscript is still very technical (therefore difficult to read) and is still lacking a broader context.

Why are the ice nucleation results still in the revised manuscript? Two reviewers correctly pointed out they are not relevant to this study, and after reading the revised manuscript they are still irrelevant and misleading. The INP results are mentioned right below the title of the section “Correlation of cell and VLP counts with environmental parameters” What

does the INP results have to do with correlations of cells and VLP counts to environmental parameters?

How was the correlation analysis to environmental factors performed? Light means photosynthetic available radiation? Salinity was measured with? What depth? Wind speed was averaged or they took the instantaneous values during measurement?

What is the purpose of the EF analysis in the context of this manuscript? It has been previously shown that the SML is enriched in comparison with the underlying water, so that is not a new result. Why not relate it to environmental variables? Maybe at different air, water temperatures, wind speeds the EF changes? Or the relationship to either film or jet drops, the primary mechanism of sea spray into the atmosphere? This section is a good example of the lack of a greater context of the manuscript; the authors just state the EFs.

I'm not sure that the loading conditions analysis, by itself, is sufficient to track the potential sources, given that the clouds might have precipitated before reaching their measuring site. I would recommend the authors to use the AIRS Precipitation Estimate data to see if the air masses they are tracking precipitated or not before they reached their sampling site. The loading analysis only tells them about a possible injection of aerosols from the boundary layer.

The authors answer to one of the reviewers that snow is another sort of precipitation and that all rain originating from raining clouds was initially in a frozen state (like snow); they why such a striking difference in the relative abundance shown in Fig. 2C in comparison to rain? Can the authors elaborate on this interesting finding?

About the flow cytometer analysis, Fig. S13. There is no explanation of panel G in the figure description. What was the noise threshold for the VLP? Panel G doesn't form a clear population and from the plot it doesn't support it is only viruses. Is there a more recent protocol than Brussaard et al. 2010?

In the discussion, line 374, the authors write that they assume the troposphere contains its own viral community. The study does not contain sufficient evidence for this statement. The

viruses could have been aerosolized in another region.

For Fig. S13 A,B, how was the threshold decided, where to gate between high and low green fluorescence?

Fig. S8, it's not accurate to say that the virus origin is aerosol or rain.

Fig. S2. Which points are foam, SML, and SSW? Like this, I cannot judge if there is a gradient towards the atmosphere.

Line 108: it should say "To fill some of these knowledge gaps,"

Line 126: Marine what?

Line 135: If VLPs in sea foams often adhere to particulate matter, aren't they less likely to aerosolize?

Line 141: across the five precipitation samples 2.3×10^3 cell ml⁻¹ was the minimum, so at this concentration there wasn't enough DNA to sequence?

Line 169: neuston and plankton should be "neuston and SSW plankton"

Lines 176-183: One model shows significance but others don't, then?

Figure 2D: I think the scale can be changed to 0-60% or even 0-40% to show more clearly the different relative abundance heat map.

[Editorial Note: Reviewer #4 was asked to assess the reply provided to Reviewer #1's comments.]

Looking back into the answers to reviewer#1:

Comment number:

1. Addressed
2. Not completely, they can relate EF to meteorological conditions to what affects or not the EFs presented
3. Answered
4. Not well addressed. The figure they show is not clear. Length of each event, amount of rain, DNA yield, should be clearly stated on a table. I believe I mentioned that on my report
5. Addressed
6. It is not clear to me why the authors choose tellurium, it is a very rare pollutant. Also, the phrase they added is not clear and quite speculative.
7. I do not think the phrase "constant genetic inflow to the sea surface by precipitation" should be used as the previous reviewer mentioned. With the data presented, it is not really possible to know where and when the viruses were injected into the atmosphere.

Minor comments:

1. Yes
2. Yes
3. Yes
4. Ok
5. This is one of my main problems. This implies, as the title does, that there are viruses unique to rain, while the authors show genetic adaptation it is far away from *proving* that the viruses originate from rain

IN data:

I would recommend removing the IN data from the manuscript. Their assay cannot in any way identify the direct ice-nucleating agent, it only gives a general INP of the filtrate, thus I don't see how it contributes any valuable information to the context and message of the current manuscript

VLP assessment via microscopy (reviewer #2, point 1):

From what I can observe from the pictures the authors uploaded, besides the foam samples, no aggregation seems to be seen, but at the same time the authors did not address the issues with fluorescent dyes in EPM in counting VLP (Forterre et al. Fake virus particles generated by fluorescence microscopy. Trends Microbiol. 2013).

Editorial Note: Mentions of prior referee reports have been redacted. Figures have been redacted as indicated to remove third-party material where no permission to publish could be obtained.

Point to point answers for reviewers

We use a color code to discriminate between the newly raised reviewer comments, given in black, the original comments in case there was still a need for discussion (also given in black, preceded by “Original related comment”), our formerly given answer (in blue) and the new answer (in dark yellow). Line numbers refer to the track change version of the manuscript.

Major changes that were made (apart from text revisions):

- We added the filtered water volumes to Table S14
- We ran another tool (PhaGCN2.0) to determine more recent taxonomic information about the viruses (now added to Table S6)
- New Table S12 about precipitation sample details
- Revision of Fig S12, S13, S8, Fig. 2D (scale) and 5 (legend name)

Looking back into the answers to reviewer#1:

Comment number:

1. Addressed
2. Not completely, they can relate EF to meteorological conditions to what affects or not the EFs presented

[REDACTED]

Previous answer: To our knowledge, the residence time of VLPs in the SML is not known, and probably highly dependent on various factors, e.g., if the SML is part of a slick (enhanced surface film), the currents, the waves, light conditions. We assume that if environmental conditions favor enrichment, it should apply to both, viruses, and prokaryotes at the same time and this is probably why we observe a correlation of EFs here. Also of course because some viruses will be attached to or replicating within bacteria at the time of fixation.

New answer 1: Enrichment factors were already correlated to meteorological conditions and applied in linear models in all previous versions of the manuscript. The data is presented in Table S2. The correlations were all not significant except for one linear model, which was mentioned in lines 180-183:

One linear model considering the combinatory effects of wind speed and salinity on the EF of small phototrophic eukaryotes in the SML was significant (F-test, $F = 5.43$, $p = 0.038$, $df = 6$), and in total 59.6 % of the residuals could be explained by this model.

We have further added to the methods in lines 507-509:

Enrichment factors (EFs) as a standard parameter in the research field were calculated as previously performed²⁵, but do not consider that the residence time of cells and VLPs in SML is unclear.

3. Answered
4. Not well addressed. The figure they show is not clear. Length of each event, amount of rain, DNA yield, should be clearly stated on a table. I believe I mentioned that on my report

[REDACTED]

Previous answer: We thank the reviewer for this comment. Details on the rain events can be found in Supplement material Fig S12, which we revised slightly and explained a bit better in the

caption. We had to pool DNA for the periods 07-09. Feb and 14th to 22nd of Feb. 2020 because otherwise the DNA yield would have been insufficient to sequence. Samples from an additional rain period from 9th to 10th Feb were too low in DNA to be sequenced. An additional rain sample from the 26th Feb. only yielded 35 mL of rain and had too little DNA to be sequenced as well.

New answer 2: Fig. S12 has been revised again, and we added Table S12 with the other information requested.

5. Addressed
6. It is not clear to me why the authors choose tellurium, it is a very rare pollutant. Also, the phrase they added is not clear and quite speculative.

[REDACTED]

Previous answer: We agree, this would be interesting to show. We also agree that the virus itself will not make use of the genes but probably its (atmospheric) host has use for them, and the virus will benefit from more enhanced replication success. For now, it just gave us confidence to see that these viruses are genetically different (high G/C, specific genes) from the seawater viruses, because these features could be an asset for enduring in atmospheric environments as high G/C, for instance, is known for bacterial isolates from stratosphere as we mention in the discussion. We discuss this now in line 367-371:

A rain-specific VC was identified, and many airborne viruses were previously unknown. Some viruses carried genes, e.g., for a photoreceptor or Tellurium resistance, the latter potentially being relevant for tolerating Tellurium contaminations in the atmosphere^[1], possibly benefiting the host after viral infection and favoring viral replication success. However, further experimental evidence is required to confirm that these genes are expressed.

New answer 3: Viruses provide auxiliary metabolic genes to their hosts. These AMGs are often somewhat ecosystem-specific, e.g. in polar regions, we find viruses that provide hosts with genes enabling survival in the cold (see <https://journals.asm.org/doi/10.1128/mSystems.00246-20>). Hence, we were looking for evidence that these viruses belong to the troposphere by looking at their AMGs and detected and name the genes for Tellurium resistance and photoreceptors because they tell us something about the possible origin of the viruses.

Basically, all descriptions of AMGs from metagenome-derived viruses in the literature are speculative until we have a viral isolate, with which we can get experimental proof. However, to date there is not a single phage isolate from the atmosphere to the best of our knowledge, so we must work with the sequence data. And even here, the number of genomes from uncultured viruses from air is < 700 in IMG/VR database, compared to 10⁷ aquatic viral genomes, so the data we have here really make an important contribution.

G/C content in bacteria from the stratosphere was equally high as in our viruses (Ellington et al., 2021), and it has recently been shown that viruses and host share a similar G/C content among all kingdoms (Simon et al, 2021), which makes it likely that the rain viruses belong to hosts of similar high G/C content. We hope this clarifies our approach.

Nevertheless, we now have removed the sentence on the Tellurium and photoreceptor genes from the discussion as the reviewer found it was too speculative.

[FIGURE REDACTED]

Figure from Simon 2021

Ellington AJ, Bryan NC, Christner BC, Reisch CR. Draft genome sequences of actinobacterial and betaproteobacterial strains isolated from the stratosphere. *Microbiol Resour Announc* 10, e0100921 (2021).

Simon D, Cristina J, Musto H (2021) Nucleotide Composition and Codon Usage Across Viruses and Their Respective Hosts. *Front Microbiol* 12:646300. doi:10.3389/fmicb.2021.646300

7. I do not think the phrase “constant genetic inflow to the sea surface by precipitation” should be used as the previous reviewer mentioned. With the data presented, it is not really possible to know where and when the viruses were injected into the atmosphere.

[REDACTED]

Previous answer: Here we refer to the high G/C viruses that were solely detected in rainwater. They could not be detected in any of the seawater samples at all, but we see that seawater prokaryotes carry CRISPR spacers against these rain viruses. Possessing these spacers means at least, that the rain viruses tried to infect the seawater prokaryotes at a certain time, though not necessarily successful. If these rain viruses do not occur in the sea, constant genetic inflow from precipitation is needed to establish an infection history as we detected by spacer-to-protospacer matches, which is what we meant with this sentence. We revised/added accordingly in line 355-357:

Established adaptive immunity indicates previous virus-host encounters, requiring a constant genetic inflow to the sea surface by precipitation, especially if viruses are not naturally marine.

New answer 4: We have removed the sentence about the constant genetic inflow. New sentence is in lines 378-381:

Such established adaptive immunity indicates previous virus-host encounters along with viruses from the atmosphere leaving their signatures in the form of host-acquired CRISPR spacers in the sea surface and suggests that viruses are probably still able to inject their genome into the host after deposition.

Minor comments:

1. Yes

2. Yes
3. Yes
4. Ok
5. This is one of my main problems. This implies, as the title does, that there are viruses unique to rain, while the authors show genetic adaptation it is far away from *proving* that the viruses originate from rain

[REDACTED]

Previous answer 13: We would be interested to know why the viruses should clearly originate from somewhere else (where from)? The viruses we report here were only detected in rain samples. Since they can still occur somewhere else (e.g. other sorts of precipitation or places that were filled up with rain they occur in), we now have rephrased to “only detected in rain” throughout the manuscript.

New answer 5: We think, this basically refers to any sample ever taken. If we take a marine sample, we assume the microbes we detect are from the sea, but in fact they could originate from everywhere else, since everything is connected, and microbes get dispersed between ecosystems. Hence, we assume that the reviewer refers to problematic wording and stepped back from using the word “originate” throughout the manuscript. The wording “unique to rain” has already been corrected/omitted in the version the reviewer saw and rephrased to “only in rain”. That ““Rain_only” refers to viral genomes exclusively found in rain **in this study**” was the explanation for this term in the caption of Figure 5, which in no way excludes that viruses could not appear in any other samples as well. We now always added “in this study” or “in the present investigation” where we referred to viruses, which were found only in rain.

IN data:

I would recommend removing the IN data from the manuscript. Their assay cannot in any way identify the direct ice-nucleating agent, it only gives a general INP of the filtrate, thus I don't see how it contributes any valuable information to the context and message of the current manuscript

Answer 6: INP concentration was determined from the 5 µm filter, not from a filtrate. The INP concentration differences between foam, SML and SSW are a part of the dataset and valid data, and the ice activity is in a range previously reported. We thus like to keep the data in the supplement and refer to them briefly in the text, now in its own paragraph in lines 190-196:

Ice nucleation activity of marine samples was highest in sea foams

The highest ice nucleation activity concluded from INP concentrations over the detectable temperature range in our samples was determined for sea foams, followed by SML and SSW samples (Supplementary methods, Fig. S5). Ice nucleation activity for all samples generally started at high temperatures of ~ -4 to -6 °C, comparable to observations for microorganisms in the atmosphere⁶⁰.

VLP assessment via microscopy (reviewer #2, point 1):

From what I can observe from the pictures the authors uploaded, besides the foam samples, no aggregation seems to be seen, but at the same time the authors did not address the issues with fluorescent dyes in EPM in counting VLP (Forterre et al. Fake virus particles generated by fluorescence microscopy. Trends Microbiol. 2013).

Answer 7: We thank the reviewer for pointing out the challenges associated with quantifying viruses regarding false positive results. As in epifluorescence microscopy (EfM), flow cytometry based on using fluorescent dyes can produce such FP results. This is one of the reasons why we also calculate enrichment factors (EFs), namely, to compare ratios derived from two samples that were treated with the same method and should have a comparable quantitative bias. Considering the problem with FPs, still quantifying viruses using EfM and flow cytometry is the state-of-the-art procedure for marine samples and already a more sophisticated method than virus enumeration by TEM or plaque assays. However, it is valuable to point out the issue. We thus have added the following sentence to the discussion to account for the possibility of fake VLPs in lines 368-369:

On the other hand, methods based on fluorescence dyes are prone to generate fake VLPs⁶⁶, which could lead to counting of false positives.

Reviewer #3 (Remarks to the Author):

The authors addressed most of my comments, and I think the “flow” of the manuscript at its present form is much improved.

Still, I cannot agree with reply 32, and some clarifications are needed:

Why did the authors provide back trajectories only for the rain samples? Is there a logic behind not providing this information also for the air samples? As the author indicate in their conclusions (L. 378): “...the air mass trajectory is crucial for understanding airborne microbial diversity and viral biogeography, being especially relevant for the highly influenced ocean-atmosphere interface.”

If no special reason for the lack of this analysis, please provide air mass trajectories on the air samples as well.

Answer 8: There is a logic behind providing trajectories (TJs) for the rain samples, but not for the aerosol samples: The clouds delivering rain have likely travelled many kilometers at an elevation well above the surface, and since we find a specific viral/bacterial community (the high G/C viruses) in the precipitation samples, the idea was to see how much chance there was for loading with marine species, i.e., how much time the related air masses + clouds previously spent over the ocean. We choose to calculate our TJs from a starting height of 700 m based on the experimental observation and model analysis highlighted in Fig S12. The starting height choice is linked to the experimental measurements, which certify (Fig S12) that the clouds (the virus/microbe vehicles) are at that altitude.

We made the restrictive choice of calculating TJs only for rain events (wet deposition) because during these events we are certain that there is a link (the precipitation) between our virus/microbe vehicles (the clouds) in atmosphere and the sampling point at the ground. In this way is it possible to connect the sampled virus with its source area through the pathway of the vector calculated back in time. On the other hand, air samples at low heights (2 m) rather represent atmospheric concentrations in the boundary layer close to the ground. However, a clear space/time link between these measurements made at the ground and the atmospheric conditions above the site might not be given. This is even more true if (as in our case) the sample is an integration over several days and then could be the expression of different meteorological conditions and air masses. Hence, concerning the aerosol samples, we do not feel it makes much sense to look at the origin of air masses for aerosols collected few meters above the ground. It is commonly known that it bears high uncertainty to run the TJs from very low heights close to the ground. This is because the meteorological models that initialize the TJs are not spatially defined enough to consider the orography and all the complex interactions between the surface and the lowest atmospheric layers. Clearly the closer the TJs are to the ground, the higher is their uncertainty.

Moreover, these problems can be accentuated in coastal areas where there may be considerable orographic gradient, which is often biased in the model representation.

Reviewer #4 (Remarks to the Author):

The manuscript "Heads in the clouds: Marine viruses disperse bidirectionally along the natural water cycle" by Rahlff et al., compared metagenomics analysis of measurements performed at 1m depth water, the SML, and foams off the shore in the bay of Tjärnö, Sweden, and aerosol and rain collected on a station located at the shore, and use these analyses to describe exchange between the ocean surface, aerosol particles and rain. While their metagenomic analysis is very well done and robust, I have many constraints on their conclusions and the methodology. After reading the revised manuscript and the answer to the comments of the previous reviewers, I cannot endorse publication of the article in the present form, the article needs major corrections. I hope my comments are helpful to the authors.

Answer 9: We thank the reviewer for the positive assessment of our analysis.

The way the article is framed underlies a misunderstanding of cloud and rain formation. This was highlighted to me in the answer to one of the reviewer's comments. A previous reviewer correctly requested, as a minor comment, to not use the term "unique to rain", and the authors responded by asking "We would be interested to know why the viruses should clearly originate from somewhere else (where from)? The viruses we report here were only detected in rain samples." This quite unfortunate answer reflects the lack of understanding of cloud and rain formation. Clouds in our atmosphere cannot form without aerosols, for water to change phase homogeneously it needs about 400% supersaturation values. These values are impossible to get in our atmosphere; therefore, a nucleolus/surface (i.e., aerosols) is needed for the water vapor to condense. Aerosols are either emitted from the surface (primary aerosols) or are formed directly in the atmosphere (secondary aerosols). Secondary aerosols are produced from the oxidation of volatile/semi-volatile organic compounds, and when they are formed they produce aerosol sizes on the order of 20nm or slightly bigger. Given that viruses do not form by oxidation and are much larger in size, they have to originate as primary aerosols, from what part of the planet and whether they are attached to another particle (e.g., bacteria) or "free-living" is a very interesting question.

Also, the authors mentioned in their introduction that viruses (and bacteria) were already found in clouds and that they might trigger their own precipitation, so we know viruses exists in clouds and precipitation, but we don't know which ones, we don't know if they can "live" in the atmosphere, we don't know how far they can be dispersed, we don't know if they can adapt to atmospheric conditions. These are questions the authors can advance our knowledge in.

Answer 10: We agree that viruses do not form from the gas phase in the atmosphere. For microorganisms spending time in different compartments like the ocean, SML, atmosphere, precipitation and then maybe also land, it is difficult to define the "origin". We think, this basically refers to any such sample ever taken, since everything is connected, and microbes get dispersed between ecosystems. Hence, we assume that the reviewer refers to a problematic wording which we used and stepped back from using the word "originate" throughout the manuscript. The wording "unique to rain" has already been corrected/omitted in the version the reviewer saw and rephrased to "only in rain". That "'Rain_only" refers to viral genomes exclusively found in rain **in this study**" was the explanation for this term in the caption of Figure 5, which in no way excludes that viruses could not appear in any other samples as well. We now

always added “in this study” or “in the present investigation” where we referred to viruses which we found only in rain.

Another constraint I have is there aerosol measurements; they are not atmospherically relevant. Aerosols collected at 2-m above the surface next to a hut are irrelevant to the boundary layer and even less to cloud processes. Those aerosols measurements are only relevant for microorganism diversity to their immediate surroundings. To be able to relate aerosols that might reach the cloud condensation level they need to be taken at least 10m above the surface. So in the context of the message of this article, those measurements cannot be used.

Answer 11: From earlier studies, we knew that cloud condensation nuclei and INP were both found in aerosols derived from SML or even in the SML itself (Christiansen et al 2020, Hendrickson et al 2021, Wagner et al. 2021, Wilson et al., 2015). In our study, we also found INP in the SML and foams, in agreement with these earlier studies. Hence, we were interested to also look at aerosol samples. We agree that conditions for our aerosol sampling were not optimal to obtain a sample representative for the local atmospheric aerosol (where, nevertheless, also a height of 10 m may or may not be sufficient, depending on the local circumstances). However, with these samples we could qualitatively demonstrate an overlap of viral and bacterial species between sea, aerosols, and rain. So overall, our aerosol samples suggest that there were emissions of viral and bacterial species from the ocean, which then reached our measurement site, and with this these samples added additional information to our study.

We added the following sentence to our methods to openly address the sampling height in relation to the intentions of the study in lines 450-455, underlined is new:

We used a land-based aerosol pump/constant flow sampler (QB1, Dadolab, Milan, Italy) with a custom-made filtration unit (SIMA-tec GmbH, Schwalmtal, Germany) to filter aerosols from the atmosphere in coastal proximity about ~2 m over ground between buildings (Fig. S11). This height is not relevant for a characterization of atmospheric aerosols including cloud condensation nuclei and INP but does provide information on seaborne aerosols and their role as viral and microbial vehicles.

References:

Hendrickson, Brianna N., Sarah D. Brooks, Daniel CO Thornton, Richard H. Moore, Ewan Crosbie, Luke D. Ziemba, Craig A. Carlson, Nicholas Baetge, Jessica A. Mirrielees, and Alyssa N. Alsante. "Role of sea surface microlayer properties in cloud formation." *Frontiers in Marine Science* 7 (2021): 596225.

Wagner, R., Ickes, L., Bertram, A.K., Els, N., Gorokhova, E., Möhler, O., Murray, B.J., Umo, N.S. and Salter, M.E., 2021. Heterogeneous ice nucleation ability of aerosol particles generated from Arctic sea surface microlayer and surface seawater samples at cirrus temperatures. *Atmospheric Chemistry and Physics*, 21(18), pp.13903-13930.

Wilson, Theodore W., Luis A. Ladino, Peter A. Alpert, Mark N. Breckels, Ian M. Brooks, Jo Browse, Susannah M. Burrows et al. "A marine biogenic source of atmospheric ice-nucleating particles." *Nature* 525, no. 7568 (2015): 234-238.

Christiansen, Sigurd, Luisa Ickes, Ines Bulatovic, Caroline Leck, Benjamin J. Murray, Allan K. Bertram, Robert Wagner et al. "Influence of Arctic Microlayers and Algal Cultures on Sea Spray

Hygroscopicity and the Possible Implications for Mixed-Phase Clouds." *Journal of Geophysical Research: Atmospheres* 125, no. 19 (2020): e2020JD032808.

Why did the authors neglect two of the four rain events into their discussion? Given that half of your measurements did not contain enough DNA to sequence is a result in itself. How much rain water was collected in each rain event? In Fig. S12 it looks that one of the rain events (9-10 feb) that did not have enough DNA was larger than one (7-9 feb) that did have enough DNA. By the way, the figure is difficult to read. ´

Answer 12: We added the rain volumes, mean precipitation, DNA concentration and info on DNA pooling to the new Table S12. The figure S12 has been revised to improve readability. The reason why we did not want to pool the rain from 7th to 9th Feb. with another sample was that there was Storm Ciara (https://en.wikipedia.org/wiki/Storm_Ciara), an extratropical cyclone over the Atlantic towards Sweden during this time, and we thought that this as a special event should be kept on a separate filter. Regarding the 9-10 Feb. sample, the reviewer is right. We did not sequence this one as we had some limitations on the sequencing plate and had to select samples. One reason we deselected it and decided for the 7th to 9th Feb sample instead is, that during such a strong storm, we can no longer be sure about other terrestrial emissions which would end up in the sample. In the end, we decided for 2 rain and 2 snow samples. We added the following new text to the manuscript in lines 489-491:

*Event 1 trajectories were associated with an extratropical cyclone (Storm Ciara), which mainly affected the United Kingdom, but also crossed northern Europe*⁸⁵.

Storm Ciara, 9th Feb 2020, 11:40 am local time, Image obtained from <https://earth.nullschool.net> showing animated winds in green. The sampling of the 9-10 feb. sample started ~2 hrs after this picture.

[FIGURE REDACTED]

In the first sentence of the Discussion, the authors state that by detecting marine MAGs and viral genomes in aerosol and rain samples, their study shows that aerolization from the sea surface

took place. Why? The rain does not come from deep convective clouds that formed around the measuring site, most likely the rain comes from stratocumulus decks travelling several 10s if not hundreds of kilometers. Moreover, there is no mention of wind speeds and direction of when the SML, foam, and SSW measurements took place, so it is difficult to assess if some viruses were emitted before a rain event and maybe scavenged into the samples, or if the emitted particles from the sea surface stations could be transported in the direction of the aerosols and rain samplers. A previous reviewer even offered an opposite scenario (i.e., deposition of cloud-born – or the scavenged below-cloud airborne microbes), and again, instead of answering a very helpful and insightful comment to make the manuscript better, the authors answer with a question. How do the authors know they aerosols were not scavenged? “highlighting” in the abstract the “bidirectional route” does not answer the comment from the previous reviewer. Furthermore, bidirectional should not be used, it is a misleading word that assumes the viruses originated from the atmosphere, and as I explained in my first comment this is not physically possible.

Answer 13: We thank the reviewer for this comment. The wind speed data were already included in the previous manuscript version, see Supplement Table S9 Field Data. We fully agree that the rain likely formed in air masses that travelled a longer distance, which is exactly the reason why we looked at the trajectories for these rain events. We also fully agree that viruses could stem from clouds or below-clouds. Nevertheless, the overlap we found of viral and bacterial species between sea and rain samples suggests that they were emitted from the ocean into the atmosphere at one point of time, as the reviewer quite correctly argued in the comments above.

This is our speculation: part of the viruses (the ones with high G/C) have spent some undefined time in the atmosphere, clouds, below clouds and probably have once come from Earth. We hope to have clarified this point in lines 401-407 of the discussion:

As genetic adaptations like the nucleic acid base composition will not change within hours, for instance shortly after aerosolization, we assume that viruses could be maintained in the atmosphere for some time and further supplied by marine or terrestrial sources as shown for bacteria sampled over the major oceans^{23,80}.

Concerning the use of the word bidirectional, we do not understand the reviewer’s point. In our understanding, “bidirectional” does not say anything about the origin but only about the route, which could be both sea → air/rain and air/rain → sea (as indicated by the “bi”). Likewise, in the title “disperse bidirectionally”, bidirectionally refers to a movement/dispersion process, not an origin. We think the bidirectional transmission is very well supported by the data showing viral/bacterial overlaps between the different ecosystems based on read mapping. For instance, consider the SNP analysis of the virus #1 population: rain, air and foam virus share the same SNPs but there are additional SNPs that are only shared between the virus populations from rain/foam, rain/air and air/foam showing “individual” contacts between these paired ecosystems. We therefore like to keep “bidirectionally” in the title.

Additionally, we want to point out that Reviewer #3 was apparently okay with our answer on the “bidirectional” exchange matter.

We specify it further in the discussion in line 412:

We conclude that viruses disperse bidirectionally (from sea to air and vice versa) along the natural water cycle ...

One of the reviewers also mentioned the lack of ecological context of the manuscript, and that the text is very technical. The revised manuscript is still very technical (therefore difficult to read) and is still lacking a broader context.

Answer 14: We thank the reviewer for this comment. To let it sound less technical, we avoid naming methods/techniques and have re-phrased statistical parts in the results, e.g., in lines 201/202, 205, 209, 259, 263

We added the following few statements in the discussion to improve ecological and broader context:

Lines 369-373

Due to enrichments of viruses in rainwater over foam compared to rainwater over SML (Fig. 3D), aerosolization from foams as a “springboard” from sea to air is likely because sea foams contain high amounts of transparent exopolymer particles (TEPs)⁶⁷. These are prone to aerosolization, were found in cloud water and to absorb viruses.

Lines 378-381

Such established adaptive immunity indicates previous virus-host encounters along with viruses from the atmosphere leaving their signatures in the form of host-acquired CRISPR spacers in the sea surface and suggests that viruses are probably still able to inject their genome into the host after deposition.

Lines 394-396

Since viruses and hosts have correlating G/C base contents even across kingdoms⁷⁸, we speculate that the here described high G/C viruses could infect hosts of similar nucleotide proportion.

Lines 415-419

Rainwater is a key component of the Earth's water cycle, and studying the microorganisms and viruses present in rainwater and their dissemination along the natural water cycle can help us better understand the cycling of water, pollutants, and nutrients in the environment. This can have further implications for water resource management, agriculture, and ecosystem health.

Why are the ice nucleation results still in the revised manuscript? Two reviewers correctly pointed out they are not relevant to this study, and after reading the revised manuscript they are still irrelevant and misleading. The INP results are mentioned right below the title of the section “Correlation of cell and VLP counts with environmental parameters” What does the INP results have to do with correlations of cells and VLP counts to environmental parameters?

Answer 15: The last version of the manuscript did not include any correlations with INP. However, we mentioned the INP concentrations in the paragraph on environmental data/VLP/cell count correlation, which was indeed misleading.

As for the general answer, several different reviewers were engaged in reviews for this manuscript before and we had an exchange on that topic earlier. Upon resubmission, the status was that the editors had suggested to keep the INP data in the manuscript (but not to correlate them), which we deemed was the final decision. We acknowledge that this is an information the reviewer might not have been aware about. We do feel that these INP data are a small but important additional information which we would like to keep. Bridging the gap between disciplines is an important topic, and INP are an important timely topic for the atmospheric community. We now present the results in a separate section in lines 190-196:

Ice nucleation activity of marine samples was highest in sea foams

The highest ice nucleation activity concluded from INP concentrations over the detectable temperature range in our samples was determined for sea foams, followed by SML and SSW samples (Supplementary methods, Fig. S5). Ice nucleation activity for all samples generally started at high temperatures of ~ -4 to -6 °C, comparable to observations for microorganisms in the atmosphere⁶⁰.

How was the correlation analysis to environmental factors performed? Light means photosynthetic available radiation? Salinity was measured with? What depth? Wind speed was averaged or they took the instantaneous values during measurement?

Answer 16: The requested details were partly mentioned in the last manuscript version line 393-397, which was now extended by the underlined parts according to the reviewer's suggestions in lines 429-432:

Wind speed was measured for ~ 1 minute with a handheld VOLTCRAFT AN-10 anemometer (Conrad Electronic, Hirschau, Germany) held at 2-3 m above the sea surface and either an approximate average was reported, or a range in case of stronger variations (Table S11). Light conditions were recorded on the boat using the Galaxy Sensors smartphone application v.1.8.10. Temperature and salinity were measured at ~20 cm beneath the surface from the small boat using a portable thermosalinometer (WTWTM MultiLineTM 3420).

Please additionally refer to Table S11 for results.

What is the purpose of the EF analysis in the context of this manuscript? It has been previously shown that the SML is enriched in comparison with the underlying water, so that is not a new result. Why not relate it to environmental variables? Maybe at different air, water temperatures, wind speeds the EF changes? Or the relationship to either film or jet drops, the primary mechanism of sea spray into the atmosphere? This section is a good example of the lack of a greater context of the manuscript; the authors just state the EFs.

Answer 17: EF is commonly used to present enrichments in the SML, we agree. The concept was introduced by the advisory board GESAMP and published in this report (<http://www.gesamp.org/site/assets/files/1237/the-sea-surface-microlayer-and-its-role-in-global-change-en.pdf>). Principally, all microlayer studies use it, which is a good thing because it makes results between studies easier comparable. EFs can be profoundly different for different conditions (surface slicks, foams, different oceans, different oceanographic features sampled, different SML samplers used to name a few). It is for example a new result to have EF of VLP for foams and SML. Also, EF of bacteria in foams compared to SML were rarely reported so far. The data the reviewer asked for, were already included, and can be seen in Supplement Table S2. There we show correlations of absolute counts with each other, EF with each other, EFs and absolute counts with environmental variables (light, wind speed and salinity). Also, all details about the correlation were given (which type of correlation, t-value, df, p-value, correlation coefficient, and degree of significance).

Im not sure that the loading conditions analysis, by itself, is sufficient to track the potential sources, given that the clouds might have precipitated before reaching their measuring site. I would recommend the authors to use the AIRS Precipitation Estimate data to see if the air masses they are tracking precipitated or not before they reached their sampling site. The loading analysis only tells them about a possible injection of aerosols from the boundary layer.

Answer 18: Indeed, our analysis only suggests possible loading. A more thorough examination is beyond the scope of this study. In agreement with the editorial office, we do not need to re-do the trajectory analysis with the recommended model. We nevertheless thank the reviewer nonetheless for the valuable suggestion and keep it in mind for future applications.

The authors answer to one of the reviewers that snow is another sort of precipitation and that all rain originating from raining clouds was initially in a frozen state (like snow); they why such a striking difference in the relative abundance shown in Fig. 2C in comparison to rain? Can they authors elaborate on this interesting finding?

Answer 19: Unfortunately, we cannot do this. The reason is that we have only a very limited amount of precipitation samples in general, and only one single snow sample, which does not allow to draw any general conclusions like “snow samples have a limited diversity” etc. because we simply don’t know and cannot extrapolate from a single sample. The data show that the snow consisted of 97% cyanobacteria, while the rain was more diverse. It could be that rain caught/washed out organisms from the atmosphere on the way down, while snow did not.

About the flow cytometer analysis, Fig. S13. There is no explanation of panel G in the figure description.

Answer 20: We added the missing description, which was lost due to version conflicts. Sorry for that. Now it is written in the figure legend:

“..., gating of virus like particles (VLPs) near detection limit of the device (threshold = 550). Potential subpopulations were not considered by extra gating (G),...”

What was the noise threshold for the VLP?

Answer 21: See above. Now the threshold value is included in the description. Additionally, we changed the zoom into the cytogram (G) itself, so that the threshold is more visible. Furthermore, we added new panel H, which shows the noise of a blank measurement. In the legend it is written as follows:

“...and determination of threshold noise of stained blank (sterile buffer (TE)), which was later subtracted from the VPL samples (H).”

Panel G doesn’t form a clear population and from the plot it doesn’t support it is only viruses.

Answer 22: In our opinion, panel G showed a distinct population of viruses (and noise subtracted later) next to the detection limit of the C6 device. Potential subpopulations, here as right shoulder, were not considered by extra gating, because subpopulation did not occur frequently, and it was not the goal of the study. We were interested in the total abundance of VLPs. Here the population as histogram:

To address the noise additionally, we added an extra panel H, where a blank and the electrical noise is represented. By presenting panel H it can be seen better that only a minimal number of events represents noise, and the gating strategy is clearer.

Is there a more recent protocol than Brussaard et al. 2010?

Answer 23: The protocol of Brussaard et al. (2010) is the most used and a standard protocol for this type of measurement. It was designed to quantify a wide variety of VLPs in marine samples and describes the method and each step very detailed. The paper represents the further development of previous method papers, has proved its applicability for a wide variety of different samples of marine origin, and is still state-of-the-art. Recent studies quantifying viruses still refer to Brussaard et al. 2010 or Marie et al. 1999, or the predecessor of the Brussaard paper, but these predecessor studies give no detailed description of the single steps. The predecessor studies of Brussaard et al. (2010) for this method to target marine viruses would be:

Marie Dominique, Brussaard Corina P. D, Thyraug Runar, Bratbak Gunnar, Vaultot Daniel. Enumeration of marine viruses in culture and natural samples by flow cytometry. *Appl. Environ. Microbiol.* 1999;65(1):45–52.

Brussaard Corina P. D, Marie Dominique, Bratbak Gunnar. Flow cytometric detection of viruses. *J. Virol. Methods.* 2000; 85:175–182.

Brussaard Corina P. D. Optimization of procedures of counting viruses by flow cytometry. *Appl. Environ. Microbiol.* 2004; 70(3):1506–1513.

For Fig. S13 A,B, how was the threshold decided, where to gate between high and low green fluorescence?

Answer 24: The threshold was set to 900 to the FL1 green exclude electrical noise. By using the histogram of the population, the minimum between the main populations represents the edge between high nucleic acid content bacteria and low nucleic acid content bacteria. This was almost stable between the samples, but was checked and additionally regated manually, if necessary.

In the discussion, line 374, the authors write that they assume the troposphere contains its own viral community. The study does not contain sufficient evidence for this statement. The viruses

could have been aerosolized in another region.

Answer 25: We agree that this is a speculation and re-phrased to leave room for alternative explanations, line 401-409:

As genetic adaptations like the nucleic acid base composition will not change within hours, for instance shortly after aerosolization, we assume that viruses could be maintained in the atmosphere for some time and further supplied by marine or terrestrial sources as shown for bacteria sampled over the major oceans^{23,80}. Alternatively, the viruses could have been derived from an unknown source (non-local, marine, or terrestrial) and could have been dispersed into the rainwater, or the rain scavenged biological material from the atmosphere on the way to Earth.

Fig. S8, it's not accurate to say that the virus origin is aerosol or rain.

Answer 26: Instead of habitat/virus origin, we call the legend heading now "sample of virus assembly"

We would like to add that this issue probably applies to all environmental samples ever sequenced: no one knows how many viruses/bacteria really belong to the sampled ecosystem or were previously introduced from somewhere else.

Fig. S2. Which points are foam, SML, and SSW? Like this, I cannot judge if there is a gradient towards the atmosphere.

Answer 27: The in-text reference to Fig S2 referred to the microscopic analysis, not to the gradient. Fig. S2 is to show that counts derived from flow cytometry and EFM correlate and does only include representative samples, as can be seen from the numbers of observations mentioned in the caption. All absolute counts showing that VLPs are highest in foam followed by SML and by SSW can be seen in Figure S1 and in Table S1. We agree that the sentence as phrased was a bit misleading and now re-phrased in lines 131-137 to:

Across all stations, viral abundance ranged between 5.0×10^7 - 1.8×10^8 , 1.3×10^7 - 3.4×10^7 and 1.4×10^7 - 2.0×10^7 VLPs mL⁻¹ in floating sea foams, the SML and SSW, respectively, supporting a VLP gradient towards the atmosphere (Table S1, Fig. S1). Numbers of VLPs were verified by microscopic analysis as shown representatively for station 4 (Fig. 1B a-c, Fig. S2, Supplementary methods), and the images revealed that VLPs in sea foams often adhered to particulate matter (Fig. 1B d).

We also added colors for different ecosystems to the points in Fig S2 as suggested by the reviewer.

Line 108: it should say "To fill some of these knowledge gaps,"

Answer 28: Has been corrected.

Line 126: Marine what?

Answer 29: We now added hyphens to show that marine refers to samples in line 127:

Marine- , aerosol-, and rain samples were collected around Tjärnö Marine Laboratory

Line 135: If VLPs in sea foams often adhere to particulate matter, aren't they less likely to aerosolize?

Answer 30: Not necessarily. As can be seen in Figure 4A and S7A, detection of viruses in foams is more correlated with detection in Aerosols, than for instance viruses from SML or SSW. Also, EFs in rain over foams were higher than EFs of rain/SML. Foams are full of TEPs, to which viruses adhere, and TEPs get aerosolized. We had this in an earlier version of the manuscript in the discussion (before we trimmed it), and now added a similar sentence again in lines 369-373:

Due to enrichments of viruses in rainwater over foam compared to rainwater over SML (Fig. 3D), aerosolization from foams as a "springboard" from sea to air is likely because sea foams contain high amounts of transparent exopolymer particles (TEPs)⁶⁷. These are prone to aerosolization, were found in cloud water and to absorb viruses.

Line 141: across the five precipitation samples 2.3×10^3 cell ml⁻¹ was the minimum, so at this concentration there wasn't enough DNA to sequence?

Answer 31: We noticed there was a typo in the text, the minimum concentration was 2.7×10^3 cells ml⁻¹. The concentration of almost all rain samples was too low to be measured and despite this we sent most of them for sequencing. The minimum concentration of prok. cells was from the snow/rain sample of 35 mL, the DNA concentration could not be determined with Qubit high sensitivity Kit, and the sequencing of the sample was impossible because the concentration for the library could not be determined. All these details can now be found in Table S12.

Line 169: neuston and plankton should be "neuston and SSW plankton"

Answer 32: Has been corrected.

Lines 176-183: One model shows significance but others don't, then?

Answer 33: We assume this comment refers to some meaning for the significant model. We have added in lines 183-185:

This could indicate that due to their bigger sizes, the enrichment of phototrophic eukaryotes in the SML is more affected by wind and currents than that of the prokaryotes.

Figure 2D: I think the scale can be changed to 0-60% or even 0-40% to show more clearly the different relative abundance heat map.

Answer 34: Has been adjusted to 0 – 60%.

REVIEWER COMMENTS

Reviewer #4 (Remarks to the Author):

I would like to thank the authors for addressing most of my comments, but I cannot endorse the article for publication in its present form. While the authors addressed my comment about the aerosol measurements taken at 2m above ground level and next to a hut in the answers to the reviewers' file and added a disclaimer to the methods, the manuscript is still written as if those aerosol measurements are atmospherically relevant; and they are not, they represent the local, and only, local aerosols present around the sampling site.

The authors wrote: "This height is not relevant for a characterization of atmospheric aerosols including cloud condensation nuclei and INP but does provide information on seaborne aerosols and their role as viral and microbial vehicles"

How does it provide information on seaborne aerosols and their role as viral and microbial vehicles?

In the abstract, the authors write:

"Virus enrichment in the 1-mm thin surface microlayer and sea foams happened selectively, and variant analysis proved virus transfer to aerosols and rain. Viruses detected in rain and aerosols showed a significantly higher percent G/C base content compared to marine viruses, likely supporting that those viruses have some atmospheric residence time allowing to be transported over long distances.... Our findings on aerosolization, adaptations, and dispersal support transmission of viruses along the natural water cycle with implications for altered carbon turnover and microbial community re-structuring in remote areas."

These phrases are misleading: as if the aerosol measurements that were made represented the aerosols in the atmospheric boundary layer, and the ones presented in this study don't.

The sections "Aerosolization of biota and decreasing diversity from marine towards atmospheric ecosystems" and "Viral diversity and transfer from the sea surface to aerosols and rain " are also written like this.

The title of the section: “Rain and aerosol viruses show adaptations toward atmospheric residence and are targeted by marine prokaryote adaptive immunity” implies long-range transport and the authors even argue in one of their replies to another reviewer that they did not do back trajectories precisely because of the height of the aerosol measurements.

On another note, in the discussion, the authors wrote: “Due to enrichments of viruses in rainwater over foam compared to rainwater over SML (Fig. 3D), aerosolization from foams as a “springboard” from sea to air is likely because sea foams contain high amounts of transparent exopolymer particles (TEPs)⁶⁷.”

I do not follow the logic of this. Why does the enrichment of viruses in rainwater over foam compared to rainwater over SML implies that aerosolization from foams is a “springboard” and that this is because of high amounts of TEP? What study shows that higher TEP concentrations cause greater aerosolization? TEP will definitely contribute to the organic matter being emitted to the atmosphere, but I'm not sure that it enhances aerosolization. I would think that, if anything, it will cause the opposite. For example, Robinson et al. 2019 showed that the formation of breaking waves, higher wind speeds, could be an effective transport and formation mechanism for TEPs to the ocean surface.

Why do they reference an article about the effect TEP has on sea bass physiological performances and survival?

Robinson, T. B., Wurl, O., Bahlmann, E., Juergens, K., and Stolle, C.: Rising bubbles enhance the gelatinous nature of the air-sea interface, *Limnol. Oceanogr.*, 64, 2358–2372, <https://doi.org/10.1002/lno.11188>, 2019.

REVIEWER COMMENTS

We thank the reviewer once more for the thorough evaluation of our manuscript. Below our answers in blue, line numbers refer to the track change version of the manuscript.

Reviewer #4 (Remarks to the Author):

I would like to thank the authors for addressing most of my comments, but I cannot endorse the article for publication in its present form. While the authors addressed my comment about the aerosol measurements taken at 2m above ground level and next to a hut in the answers to the reviewers' file and added a disclaimer to the methods, the manuscript is still written as if those aerosol measurements are atmospherically relevant; and they are not, they represent the local, and only, local aerosols present around the sampling site.

In accordance with the editorial office, we have now replaced the term atmosphere by boundary layer, where referring to aerosols sampled at ~2m height, and boundary layer and rain, where referring to both sample types. When referring to aerosols we occasionally refer to “boundary layer aerosols” to remind the reader frequently about the low sampling height.

The authors wrote: “This height is not relevant for a characterization of atmospheric aerosols including cloud condensation nuclei and INP but does provide information on seaborne aerosols and their role as viral and microbial vehicles”
How does it provide information on seaborne aerosols and their role as viral and microbial vehicles?

We found overlap between marine, 2 m aerosols, and rain viruses + microbes as defined by read mapping and SNP analysis, and CRISPR spacer-protospacer interaction. This implies that the aerosol samples we collected were at least partially of marine origin and as such can provide information on seaborne aerosols.

In the abstract, the authors write:

“Virus enrichment in the 1-mm thin surface microlayer and sea foams happened selectively, and variant analysis proved virus transfer to aerosols and rain. Viruses detected in rain and aerosols showed a significantly higher percent G/C base content compared to marine viruses, likely supporting that those viruses have some atmospheric residence time allowing to be transported over long distances.... Our

findings on aerosolization, adaptations, and dispersal support transmission of viruses along the natural water cycle with implications for altered carbon turnover and microbial community re-structuring in remote areas.”

These phrases are misleading: as if the aerosol measurements that were made represented the aerosols in the atmospheric boundary layer, and the ones presented in this study don't.

In accordance with the editorial office, the two here underlined phrases have been deleted from the abstract

The sections “Aerosolization of biota and decreasing diversity from marine towards atmospheric ecosystems” and “Viral diversity and transfer from the sea surface to aerosols and rain “ are also written like this.

Section title now reads: Aerosolization of biota and decreasing diversity from marine ecosystems towards the boundary layer

Word “atmospheric” in this section has been replaced by “rain and aerosol”

The section: Viral diversity and transfer from the sea surface to aerosols and rain does not say anything about height or relevance of aerosols, it states numbers, differences, and overlaps, which are based on facts. Rain stems from the condensation and aggregation of water droplets within clouds, typically located even above the ABL and the rain will wash out suspended airborne particles from the ABL on its way to the ground. Therefore, completely neglecting that biota could stem from the ABL is not a good idea either.

Again, we removed the word atmosphere as suggested by the editorial office and replaced it by “boundary layer and rain”.

The title of the section: “Rain and aerosol viruses show adaptations toward atmospheric residence and are targeted by marine prokaryote adaptive immunity” implies long-range transport and the authors even argue in one of their replies to another reviewer that they did not do back trajectories precisely because of the height of the aerosol measurements.

Has been rephrased to: Rain and aerosol viruses show adaptations toward their ecosystems and are targeted by marine prokaryote adaptive immunity (line 327/328)

On another note, in the discussion, the authors wrote: “Due to enrichments of viruses in rainwater over foam compared to rainwater over SML (Fig. 3D),

aerosolization from foams as a “springboard” from sea to air is likely because sea foams contain high amounts of transparent exopolymer particles (TEPs)⁶⁷.”

I do not follow the logic of this. Why does the enrichment of viruses in rainwater over foam compared to rainwater over SML implies that aerosolization from foams is a “springboard” and that this is because of high amounts of TEP? What study shows that higher TEP concentrations cause greater aerosolization? TEP will definitely contribute to the organic matter being emitted to the atmosphere, but I'm not sure that it enhances aerosolization. I would think that, if anything, it will cause the opposite. For example, Robinson et al. 2019 showed that the formation of breaking waves, higher wind speeds, could be an effective transport and formation mechanism for TEPs to the ocean surface.

Why do they reference an article about the effect TEP has on sea bass physiological performances and survival?

Robinson, T. B., Wurl, O., Bahlmann, E., Juergens, K., and Stolle, C.: Rising bubbles enhance the gelatinous nature of the air-sea interface, *Limnol. Oceanogr.*, 64, 2358–2372, <https://doi.org/10.1002/lno.11188>, 2019.

We thank the reviewer for this comment and due to its speculative nature have removed the whole respective discussion paragraph.